# Variational Continual Test-Time Adaptation

## Abstract

Continual Test-Time Adaptation (CTTA) task investigates effective domain adaptation under the scenario of continuous domain shifts during testing time. Due to the utilization of solely unlabeled samples, there exists significant uncertainty in model updates, leading CTTA to encounter severe error accumulation issues. In this paper, we introduce VCoTTA, a variational Bayesian approach to measure uncertainties in CTTA. At the source stage, we transform a pretrained deterministic model into a Bayesian Neural Network (BNN) via a variational warm-up strategy, injecting uncertainties into the model. During the testing time, we employ a mean-teacher update strategy using variational inference for the student model and exponential moving average for the teacher model. Our novel approach updates the student model by combining priors from both the source and teacher models. The evidence lower bound is formulated as the cross-entropy between the student and teacher models, along with the Kullback-Leibler (KL) divergence of the prior mixture. Experimental results on three datasets demonstrate the method's effectiveness in mitigating error accumulation within the CTTA framework. Our code is anonymously available at https://anonymous.4open.science/r/vcotta-D2C3/.

## 1 Introduction

Continual Test-Time Adaptation (CTTA) [51] aims to enable a model to accommodate a sequence of distinct distribution shifts during the testing time, making it applicable to various risk-sensitive applications in open environments, such as autonomous driving and medical imaging. However, real-world non-stationary test data exhibit high uncertainty in their temporal dynamics [23], presenting challenges related to error accumulation [51]. Previous CTTA studies rely on methods that enforce prediction confidence, such as entropy minimization. However, these approaches often lead to predictions that are overly confident and less well-calibrated, thus limiting the model's ability to quantify risks during predictions. The reliable estimation of uncertainty becomes particularly crucial in the context of continual distribution shift [40]. It is meaningful to design a model capable of encoding the uncertainty associated with temporal dynamics and effectively handling distribution shifts. The objective of this paper is to devise a CTTA procedure that not only enhances predictive accuracy under distribution shifts but also provides reliable uncertainty estimates.

To address the above problem, we refer to the Bayesian Inference (BI) [1], which retains a distribution over model parameters that indicates the plausibility of different settings given the observed data, and it has been witnessed as effective in traditional continual learning tasks [38]. In Bayesian continual learning, the posterior in the last learning task is set to be the current prior which will be multiplied by the current likelihood. This kind of prior transmission is designed to reduce catastrophic forgetting in continual learning. However, this is not feasible in CTTA because unlabeled data may introduce unreliable prior. As shown in Fig. 1, an unreliable prior may lead to a poor posterior, which may then propagate errors to the next inference, leading to the accumulation of errors.

Submitted to 38th Conference on Neural Information Processing Systems (NeurIPS 2024). Do not distribute.

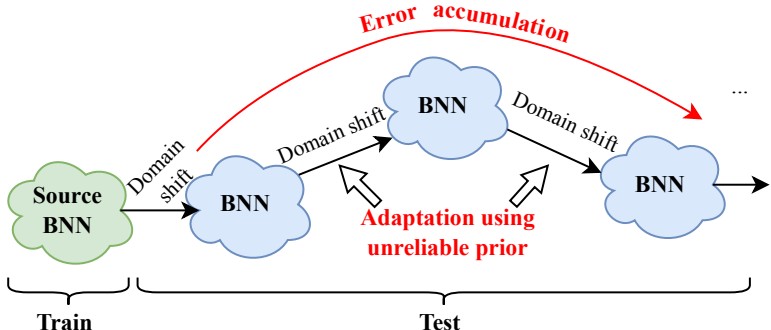

Figure 1: In CTTA task, a BNN model is first trained on a source dataset, and then is used to adapt to updated with unreliable priors, which may result in error accumulations.

Thus, we delve into the utilization of BI framework to evaluate model uncertainty in CTTA, aiming to mitigate the impact of unreliable priors and reduce the error propagation. To approximate the intractable likelihood in BI, we adopt to use online Variational Inference (VI) [49, 42], and accordingly name our method Variational Continual Test-Time Adaptation (VCoTTA). At the source stage, we first transform a pretrained deterministic model, say CNN, into a Bayesian Neural Network (BNN) by a variational warm-up strategy, where the local reparameterization trick [27] is used to inject uncertainties into the source model. During the testing phase, we employ a mean-teacher update strategy, where the student model is updated via VI and the teacher model is updated by the exponential moving average. Specifically, for the update of the student model, we propose to use a mixture of priors from both the source and teacher models, then the Evidence Lower BOund (ELBO) becomes the cross-entropy between the student and teachers plus the KL divergence of the prior mixture. We demonstrate the effectiveness of the proposed method on three datasets, and the results show that the proposed method can mitigate the error accumulation in CTTA and obtain clear performance improvements.

Our contributions are three-fold:

(1) This paper develops VCoTTA, a simple yet general framework for continual test-time adaptation that leverages online VI within BNN.

(2) We propose to transform an off-the-shelf model into a BNN via a variational warm-up strategy, which injects uncertainties into the model.

(3) We build a mean-teacher structure for CTTA, and propose a strategy to blend the teacher's prior with the source's prior to mitigate unreliable prior problem.

## 2 Related Work

### 2.1 Continual Test-Time Adaptation

Test-Time Adaptation (TTA) enables the model to dynamically adjust to the characteristics of the test data, i.e. target domain, in a source-free and online manner [25, 46, 50]. Previous works have enhanced TTA performance through the designs of unsupervised loss [37, 58, 32, 9, 7, 17]. These endeavours primarily focus on enhancing adaptation within a fixed target domain, representing a single-domain TTA setup, where models adapt to a specific target domain and then reset to their original pretrained state with the source domain, prepared for the next target domain adaptation.

Recently, CTTA [51] has been introduced to tackle TTA within a continuously changing target domain, involving long-term adaptation. This configuration often grapples with the challenge of error accumulation [47, 51]. Specifically, prolonged exposure to unsupervised loss from unlabeled test data during long-term adaptation may result in significant error accumulation. Additionally, as the model is intent on learning new knowledge, it is prone to forgetting source knowledge, which poses challenges when accurately classifying test samples similar to the source distribution.

To solve the two challenges, the majority of the existing methods focus on improving the confidence of the source model during the testing phase. These methods employ the mean-teacher architecture [47] to mitigate error accumulation, where the student learns to align with the teacher and the teacher

updates via moving average with the student. As to the challenge of forgetting source knowledge, some methods adopt augmentation-averaged predictions [51, 2, 11, 55] for the teacher model, strengthening the teacher's confidence to reduce the influence from highly out-of-distribution samples. Some methods, such as [11, 6], propose to adopt the contrastive loss to maintain the already learnt semantic information. Some methods believe that the source model is more reliable, thus they are designed to restore the source parameters [51, 2]. Though the above methods keep the model from confusion of vague pseudo labels, they may suffer from overly confident predictions that are less calibrated. To mitigate this issue, it is helpful to estimate the uncertainty in the neural network.

## 2.2 Bayesian Neural Network

Bayesian framework is natural to incorporate past knowledge and sequentially update the belief with new data [59]. The bulk of work on Bayesian deep learning has focused on scalable approximate inference methods. These methods include stochastic VI [22, 34], dropout [16, 27] and Laplace approximation [41, 15] etc., and leveraging the stochastic gradient descent (SGD) trajectory, either for a deterministic approximation or sampling. In a BNN, we specify a prior $p(\boldsymbol{\theta})$ over the neural network parameters, and compute the posterior distribution over parameters conditioned on training data, $p(\boldsymbol{\theta}|\mathcal{D}) \propto p(\boldsymbol{\theta})p(\mathcal{D}|\boldsymbol{\theta})$. This procedure should give considerable advantages for reasoning about predictive uncertainty, which is especially relevant in the small-data setting.

Crucially, when performing Bayesian inference, we need to choose a prior distribution that accurately reflects the prior beliefs about the model parameters before seeing any data [18, 14]. In conventional static machine learning, the most common choice for the prior distribution over the BNN weights is the simplest one: the isotropic Gaussian distribution. However, this choice has been proved indeed suboptimal for BNNs [14]. Recently, some studies estimate uncertainty in continual learning within a BNN framework, such as [38, 12, 13, 28]. They set the current prior to the previous posterior to mitigate catastrophic forgetting. However, the prior transmission is not reliable in the unsupervised CTTA task. Any prior mistakes will be enlarged by adaptation progress, manifesting error accumulation. To solve the unreliable prior problem, this paper proposes a prior mixture method based on VI.

## 3 Variational Inference in CTTA

We start from the supervised BI in typical continual learning, where the model aims to learn multiple classification tasks in sequence. Let $\mathcal{D} = \{(x_n, y_n)\}_{n=1}^{N}$ be the training set, where $x_n$ and $y_n$ denotes the training sample and the corresponding class label. The task $t$ is to learn a direct posterior approximation over the model parameter $\boldsymbol{\theta}$ as follows.

$$p(\boldsymbol{\theta}|\mathcal{D}_{1:t}) \propto p_t(\boldsymbol{\theta})p(\mathcal{D}_t|\boldsymbol{\theta}), \qquad (1)$$

where $p(\boldsymbol{\theta}|\mathcal{D}_{1:t})$ denotes the posterior of sequential tasks on the learned parameter and $p(\mathcal{D}_t|\boldsymbol{\theta})$ is the likelihood of the current task. The current prior $p_t(\boldsymbol{\theta})$ is regarded as the given knowledge. [38] proposes that this current prior can be the posterior learned in the last task, *i.e.*, $p_t(\boldsymbol{\theta}) = p(\boldsymbol{\theta}|\mathcal{D}_{1:t-1})$, where the inference becomes

$$p(\boldsymbol{\theta}|\mathcal{D}_{1:t}) \propto p(\boldsymbol{\theta}|\mathcal{D}_{1:t-1})p(\mathcal{D}_t|\boldsymbol{\theta}). \qquad (2)$$

The detailed process can be shown in Appendix A.

In contrast to continual learning, CTTA faces a sequence of learning tasks in test time without any label information, requiring the model to adapt to each novel domain sequentially. In this case, we assume that each domain is i.i.d. and the classes are separable following many unsupervised studies [36, 48, 5], more details about the assumption can be seen in Appendix B.1. We use $\mathcal{U} = \{x_n\}_{n=1}^{N}$ to represent the unlabeled test dataset. The CTTA model is first trained on a source dataset $\mathcal{D}_0$, and then adapted to unlabeled test domains starting from $\mathcal{U}_1$. For the $t$-th adaptation, we have

$$p(\boldsymbol{\theta}|\mathcal{U}_{1:t} \cup \mathcal{D}_0) \propto p_t(\boldsymbol{\theta})p(\mathcal{U}_t|\boldsymbol{\theta}). \qquad (3)$$

Similarly, we can set the last posterior to be the current prior, *i.e.*, $p_t(\boldsymbol{\theta}) = p(\boldsymbol{\theta}|\mathcal{U}_{1:t-1} \cup \mathcal{D}_0)$ and $p_1(\boldsymbol{\theta}) = p(\boldsymbol{\theta}|\mathcal{D}_0)$. However, employing BI for adaptation on unlabeled testing data can result in untrustworthy posterior estimates. Therefore, during subsequent adaptation, the untrustworthy posterior automatically transform into unreliable priors, leading to error accumulation. In other words,

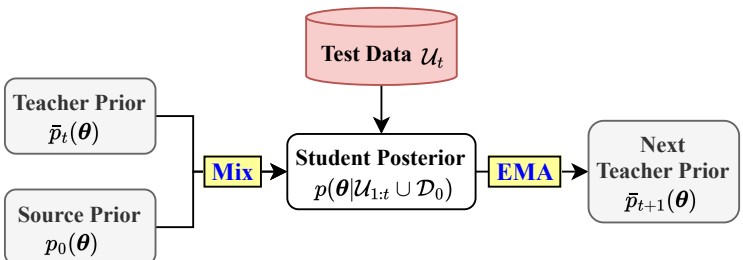

Figure 2: VCoTTA is built on mean-teacher structure, and conducts VI in CTTA using a mixture of teacher prior and source prior. The next teacher prior is updated by the exponential moving average.

124 an unreliable prior $p_t(\boldsymbol{\theta})$ will make the current posterior even less trustworthy. Moreover, the joint
125 likelihood $p(\mathcal{U}_t|\boldsymbol{\theta})$ for $t > 0$ is intractable on unlabeled data.

126 To make the BI feasible in CTTA task, in this paper, we transform the question to an easy-to-compute
127 form. Referring to [20], the unsupervised inference can be transformed into

$$p(\boldsymbol{\theta}|\mathcal{U}) \propto p(\boldsymbol{\theta}) \exp\left(-\lambda H(\mathcal{U}|\boldsymbol{\theta})\right), \tag{4}$$

128 where $H$ denotes the conditional entropy and $\lambda$ is a scalar hyperparameter to weigh the entropy term.
129 This simple form reveals that the prior belief about the conditional entropy of labels is given by the
130 inputs. The observation of the input $\mathcal{U}$ provides information on the drift of the input distribution, which
131 can be used to update the belief over the learned parameters $\boldsymbol{\theta}$ through Eq. (4). Consequently, this
132 allows the utilization of unlabeled data for BI. More detailed derivations can be seen in Appendix B.2.

133 In a BNN, the posterior distribution is often intractable and some approximation methods are required,
134 even when calculating the initial posterior. In this paper, we leverage online VI, as it typically
135 outperforms the other methods for complex models in the static setting [4]. VI defines a variational
136 distribution $q(\boldsymbol{\theta})$ to approxmiate the posterior $p(\boldsymbol{\theta}|\mathcal{U})$. The approximation process is as follows.

$$q_t(\boldsymbol{\theta}) = \arg\min_{q \in \mathbb{Q}} \mathrm{KL}\left[q(\boldsymbol{\theta}) \parallel \frac{1}{Z_t} p_t(\boldsymbol{\theta}) e^{-\lambda H(\mathcal{U}_t|\boldsymbol{\theta})}\right], \tag{5}$$

137 where $\mathbb{Q}$ is the distribution searching space and $Z_t$ is the intractable normalizing hyperparameter.
138 Thus, referring to the derivations in Appendix C, the ELBO is computed by

$$\mathrm{ELBO} = -\lambda \mathbb{E}_{\boldsymbol{\theta} \sim q(\boldsymbol{\theta})} H(\mathcal{U}_t|\boldsymbol{\theta}) - \mathrm{KL}\left(q(\boldsymbol{\theta})||p_t(\boldsymbol{\theta})\right). \tag{6}$$

139 Optimizing with Eq. (6) makes model adapt to domain shift. While VI offers a good framework
140 for measuring uncertainty in CTTA, it is noteworthy that VI does not directly address the issue of
141 unreliable priors. The error accumulation remains a significant concern.

142 Despite this, the form of the ELBO in variational inference offers a pathway for mitigating the impact
143 of unreliable priors. In Eq. (6), the *entropy term* may result in overly confident predictions that are
144 less calibrated, while the *KL term* may be directly affected by an unreliable prior. In the following
145 section, we will discuss how to solve the problems when computing the two terms.

## 4 Adaptation and Inference in VCoTTA

### 4.1 Entropy term: VI by Mean-Teacher Architecture

148 In the above section, we introduce the VI in CTTA but challenges remain, *i.e.*, the unreliable prior.
149 To mitigate the challenge in the entropy term, we adopt a Mean-Teacher (MT) structure [47] in the
150 Bayesian inference process. MT is initially proposed in semi-supervised and unsupervised learning,
151 where the teacher model guides the unlabeled data, helping the model generalize and improve
152 performance with the utilization of large-scale unlabeled data.

153 MT structure is composed of a student model and a teacher model, where the student model learns
154 from the teacher and the teacher updates using Exponential Moving Average (EMA) [24]. In VI, the
155 student is set to be the variational distribution $q(\boldsymbol{\theta})$, which is a Gaussian mean-field approximation
156 for its simplicity. It is achieved by stacking the biases and weights of the network as follows.

$$q(\boldsymbol{\theta}) = \prod_d \mathcal{N}\left(\boldsymbol{\theta}_d; \mu_d, \mathrm{diag}(\sigma_d^2)\right), \tag{7}$$

where $d$ denotes each dimension of the parameter. The teacher model $\bar{p}(\boldsymbol{\theta})$ (we use bar to distinguish the general prior) is also a Gaussian distribution. Thus, the student model is updated by aligning it with the teacher model through the use of a cross-entropy (CE) loss

$$L_{\text{CE}}(q, \bar{p}) = -\mathbb{E}_{\boldsymbol{\theta} \sim q(\boldsymbol{\theta})} \mathbb{E}_{x \sim \mathcal{U}} \left[ \bar{p}(x|\boldsymbol{\theta}) \log q(x|\boldsymbol{\theta}) \right]. \tag{8}$$

In our implementation, we also try to use Symmetric Cross-Entropy (SCE) [53] in CTTA,

$$L_{\text{SCE}}(q, \bar{p}) = -\mathbb{E}_{\boldsymbol{\theta} \sim q(\boldsymbol{\theta})} \mathbb{E}_{x \sim \mathcal{U}} \left[ \bar{p}(x|\boldsymbol{\theta}) \log q(x|\boldsymbol{\theta}) + q(x|\boldsymbol{\theta}) \log \bar{p}(x|\boldsymbol{\theta}) \right]. \tag{9}$$

SCE balances the gradient for high and low confidence, benefiting the unsupervised learning.

## 4.2 KL term: Mixture-of-Gaussian Prior

For the KL term, to reduce the impact of unreliable prior, we propose a mixing-up approach to combining the teacher and source prior adaptatively. The source prior is warmed up upon the pretrained deterministic model $p_1(\boldsymbol{\theta}) = p(\boldsymbol{\theta}|\mathcal{D}_0)$ (see Sec. 4.3.1). The teacher model $\bar{p}_t(\boldsymbol{\theta})$ is updated by EMA (see Sec. 4.3.3). We assume that the prior should be the mixture of the two Gaussian priors. Using only the source prior, the adaptation is limited. While using only the teacher prior, the prior is prone to be unreliable.

We use the mean entropy derived from a given serious data augmentation to represent the confidence of the two prior models, and mix up the two priors with a modulating factor

$$\alpha = \frac{1}{|\mathcal{I}|} \sum_{i \in \mathcal{I}} \frac{e^{H(x|\boldsymbol{\theta}_0)/\tau}}{e^{H(x|\boldsymbol{\theta}_0)/\tau} + e^{H(x|\bar{\boldsymbol{\theta}})/\tau}}, \tag{10}$$

where $\mathcal{I}$ denotes augmentation types. $\boldsymbol{\theta}_0$ and $\bar{\boldsymbol{\theta}}$ are the parameters of the source model and the teacher model. $\tau$ means the temperature factor. Thus, as shown in Fig. 3(b), the current prior $p_t(\boldsymbol{\theta})$ is set to the mixture of priors as

$$p_t(\boldsymbol{\theta}) = \alpha \cdot p_1(\boldsymbol{\theta}) + (1 - \alpha) \cdot \bar{p}_t(\boldsymbol{\theta}). \tag{11}$$

In the VI, we use the upper bound to update the KL term [31] (see Appendix D.1) for simplicity,

$$\text{KL}(q||p_t) \leq \alpha \cdot \text{KL}(q||p_0) + (1 - \alpha) \cdot \text{KL}(q||\bar{p}_t). \tag{12}$$

Furthermore, we also improve the teacher-student alignment in the entropy term (see Eq. (9)) by picking up the augmented logits with a larger confidence than the raw data. That is, we replace the teacher log-likelihood $\log \bar{p}(x|\boldsymbol{\theta})$ by

$$\log \bar{p}'(x|\boldsymbol{\theta}) = \frac{\sum_{i \in \mathcal{I}} \mathbf{1}\left(f(\bar{p}(x'_i)) > f(\bar{p}(x)) + \epsilon\right) \cdot \log \bar{p}(x'_i)}{\sum_{i \in \mathcal{I}} \mathbf{1}\left(f(\bar{p}(x'_i)) > f(\bar{p}(x)) + \epsilon\right)}, \tag{13}$$

where, for brevity, we let $\bar{p}(x'_i) = \bar{p}(x'_i|\boldsymbol{\theta})$ and $\bar{p}(x) = \bar{p}(x)|\boldsymbol{\theta})$ in short. $f(\cdot)$ is the confidence function. $\epsilon$ denotes the confidence margin and $\mathbf{1}(\cdot)$ is an indicator function. Eq. (13) can be regarded as a filter, meaning that for each sample, the reliable teacher is represented by the average of its augmentations with $\epsilon$ more confidence. In Appendix D.2, we prove that the proposed mixture-of-Gaussian is benifical to CTTA. In Appendix E.1, we discuss the influence of different $\epsilon$.

## 4.3 Adaptation and Inference

### 4.3.1 Variational Warm-up

To obtain a source BNN, instead of training a model from scratch on the source data $\mathcal{D}_0$, we transform a pretrained deterministic CNN to a BNN by variational warm-up strategy. Specifically, we leverage the local reparameterization trick [27] to add stochastic parameters, and warm up the model:

$$q_0(\boldsymbol{\theta}) = \arg \min_{q \in \mathbb{Q}} \text{KL}\left[ q(\boldsymbol{\theta}) \parallel \frac{1}{Z_0} p(\boldsymbol{\theta}) p(\mathcal{D}_0|\boldsymbol{\theta}) \right], \tag{14}$$

where $p(\boldsymbol{\theta})$ represents the prior distribution, say the pretrained deterministic model. Eq. (14) denotes a standard VI on the source data, and we optimize the ELBO to obtain the variational distribution [49]. By the variational warm-up, we can easily transform an off-the-shelf pretrained model into a BNN with a stochastic dynamic. The variational warm-up strategy is outlined in Algorithm 1.

The warm-up strategy is a common approach in TTA and CTTA tasks to further build knowledge structure for the source model, such as [26, 45, 11, 8]. Some other methods may not use warm-up but still use the source data, such as [39]. The warm-up strategy

---

**Algorithm 1** Variational warm-up

1: **Input:** Source data $\mathcal{D}_0$, pretrained model $p_0(\boldsymbol{\theta})$
2: Initialize prior distribution $p(\boldsymbol{\theta})$ with $p_0(\boldsymbol{\theta})$
3: Update $p(\boldsymbol{\theta}|\mathcal{D}_0) \approx q_0(\boldsymbol{\theta})$ by $p(\boldsymbol{\theta})$ and $\mathcal{D}_0$ using Eq. (14)
4: **Output:** Source prior $p_1(\boldsymbol{\theta}) = p(\boldsymbol{\theta}|\mathcal{D}_0)$

---

uses the source data only before deploying the model to CTTA scenario, and it is regarded as a part of pretraining. All of these methods using source data are operationalized in source-free at test time and find it is beneficial to CTTA. We use the warm-up to inject the uncertainties into a given source model, i.e., turning an off-the-shelf pretrained CNN model into a pretrained BNN model. This is convenient to obtain a pretrained BNN, because the warm-up strategy uses only a few epochs. We offer more discussions and experiments on the proposed variational warm-up strategy in Appendix F.

### 4.3.2 Student update via VI

The student model $q_t(\boldsymbol{\theta})$ is adapted by approximating using Eq. (5), and is optimized on:

$$L(q_t) = L_{\text{SCE}}(q_t, \bar{p}'_t) + \alpha \cdot \text{KL}\left(q_t || q_0\right) + (1 - \alpha) \cdot \text{KL}\left(q_t || \bar{q}_t\right), \qquad (15)$$

where $\bar{p}'_t$ is the current augmented teacher model in Eq. (13), and $p_1(\boldsymbol{\theta}) \approx q_0(\boldsymbol{\theta})$, $\bar{p}_t(\boldsymbol{\theta}) \approx \bar{q}_t(\boldsymbol{\theta})$. The KL term between two Gaussians can be computed in a closed form.

### 4.3.3 Teacher update via EMA

The teacher model is updated using EMA. Let $(\boldsymbol{\mu}, \boldsymbol{\sigma})$ and $(\bar{\boldsymbol{\mu}}, \bar{\boldsymbol{\sigma}})$ be the mean and standard deviation of the student and teacher model, respectively. At test time, the teacher model $\bar{q}_t(\boldsymbol{\theta})$ is updated by

$$\bar{\boldsymbol{\mu}} \leftarrow \beta\bar{\boldsymbol{\mu}} + (1 - \beta)\boldsymbol{\mu}, \quad \bar{\boldsymbol{\sigma}} \leftarrow \beta\bar{\boldsymbol{\sigma}} + (1 - \beta)\boldsymbol{\sigma}. \qquad (16)$$

Although the std is not used in the cross entropy to compute the likelihood, the teacher prior distribution is important to adjust the student distribution via the KL term.

### 4.3.4 Model inference

At any time, CTTA model needs to predict and adapt to the unlabeled test data. In our VCoTTA, we also use the mixed prior to serve as the inference model. That is, for a test data point $x$, the model inference is represented by

$$p_t(x) = \int p(x|\boldsymbol{\theta})p_t(\boldsymbol{\theta})d\boldsymbol{\theta} = \int \alpha p(x|\boldsymbol{\theta})p_1(\boldsymbol{\theta}) + (1 - \alpha)p(x|\boldsymbol{\theta})\bar{p}_t(\boldsymbol{\theta})d\boldsymbol{\theta}, \qquad (17)$$

For the data prediction, the model only uses the expectation to reduce the stochastic, but leverages stochastic dynamics in domain adaptation.

### 4.3.5 The algorithm

We illustrate the whole algorithm in Algorithm 2. We first transform an off-the-shelf pretrained model into BNN via the variational warm-up strategy (Sec. 4.3.1). After that, we obtain a BNN, and for each domain shift, we forward and adapt each test data point in an MT architecture. For a data point $x$, we first predict the class label using the mixture of the source model and the teacher model (Sec. 4.3.4). Then, we update the student model using VI, where we use cross entropy to compute the entropy term and use the mixture of

---

**Algorithm 2** Variational CTTA

1: **Input:** Source data $\mathcal{D}_0$, pretrained model $p_0(\boldsymbol{\theta})$, Unlabeled test data from different domain $\mathcal{U}_{1:T}$
2: $p_1(\boldsymbol{\theta}) = $ Variational warm-up$(\mathcal{D}_0, p_0(\boldsymbol{\theta}))$. // Alg. 1
3: **for** Domain shift $t = 1$ **to** $T$ **do**
4:     **for** Test data $x \sim \mathcal{U}_t$ **do**
5:         Model predict for $x$ (Eq. (17))
6:         Update student model using $x$ (Eq. (15))
7:         Update teacher model via EMA (Eq. (16))
8:     **end for**
9: **end for**

---

priors for the KL term (Sec. 4.3.2). Finally, we update the BNN teacher model via EMA (Sec. 4.3.3). See more details in Appendix G. The process is feasible for any test data without labels.

Table 1: Classification error rate (%) for the standard CIFAR10-to-CIFAR10C CTTA task. All results are evaluated with the largest corruption severity level 5 in an online fashion. C1 to C15 are 15 corruptions for the datasets (see Sec. 5.1). CIFAR100C and ImagenetC use the same setup.

| Method | C1 | C2 | C3 | C4 | C5 | C6 | C7 | C8 | C9 | C10 | C11 | C12 | C13 | C14 | C15 | Avg |
|---|---|---|---|---|---|---|---|---|---|---|---|---|---|---|---|---|
| Source | 72.3 | 65.7 | 72.9 | 46.9 | 54.3 | 34.8 | 42.0 | 25.1 | 41.3 | 26.0 | 9.3 | 46.7 | 26.6 | 58.5 | 30.3 | 43.5 |
| BN | 28.1 | 26.1 | 36.3 | 12.8 | 35.3 | 14.2 | 12.1 | 17.3 | 17.4 | 15.3 | 8.4 | 12.6 | 23.8 | 19.7 | 27.3 | 20.4 |
| Tent [50] | 24.8 | 20.6 | 28.5 | 15.1 | 31.7 | 17.0 | 15.6 | 18.3 | 18.3 | 18.1 | 11.0 | 16.8 | 23.9 | 18.6 | 23.9 | 20.1 |
| CoTTA [51] | 24.5 | 21.5 | 25.9 | 12.0 | 27.7 | 12.2 | 10.7 | 15.0 | 14.1 | 12.7 | 7.6 | 11.0 | 18.5 | 13.6 | 17.7 | 16.3 |
| RoTTA [56] | 30.3 | 25.4 | 34.6 | 18.3 | 34.0 | 14.7 | 11.0 | 16.4 | 14.6 | 14.0 | 8.0 | 12.4 | 20.3 | 16.8 | 19.4 | 19.3 |
| PETAL [2] | 23.7 | 21.4 | 26.3 | 11.8 | 28.8 | 12.4 | 10.4 | 14.8 | 13.9 | 12.6 | 7.4 | 10.6 | 18.3 | 13.1 | 17.1 | 16.2 |
| SATA [6] | 23.9 | 20.1 | 28.0 | 11.6 | 27.4 | 12.6 | 10.2 | 14.1 | 13.2 | 12.2 | 7.4 | 10.3 | 19.1 | 13.3 | 18.5 | 16.1 |
| DSS [52] | 24.1 | 21.3 | 25.4 | 11.7 | 26.9 | 12.2 | 10.5 | 14.5 | 14.1 | 12.5 | 7.8 | 10.8 | 18.0 | 13.1 | 17.3 | 16.0 |
| SWA [55] | 23.9 | 20.5 | 24.5 | 11.2 | 26.3 | 11.8 | 10.1 | 14.0 | 12.7 | 11.5 | 7.6 | 9.5 | 17.6 | 12.0 | 15.8 | 15.3 |
| VCoTTA (Ours) | **18.1** | **14.9** | **22.0** | **9.7** | **22.6** | **11.0** | **9.5** | **11.4** | **10.6** | **10.5** | **6.5** | **9.4** | **15.6** | **11.0** | **14.5** | **13.1** |

Table 2: Classification error rate (%) for the standard CIFAR100-to-CIFAR100C CTTA task.

| Method | C1 | C2 | C3 | C4 | C5 | C6 | C7 | C8 | C9 | C10 | C11 | C12 | C13 | C14 | C15 | Avg |
|---|---|---|---|---|---|---|---|---|---|---|---|---|---|---|---|---|
| Source | 73.0 | 68.0 | 39.4 | 29.3 | 54.1 | 30.8 | 28.8 | 39.5 | 45.8 | 50.3 | 29.5 | 55.1 | 37.2 | 74.7 | 41.2 | 46.4 |
| BN | 42.1 | 40.7 | 42.7 | 27.6 | 41.9 | 29.7 | 27.9 | 34.9 | 35 | 41.5 | 26.5 | 30.3 | 35.7 | 32.9 | 41.2 | 35.4 |
| Tent [50] | 37.2 | 35.8 | 41.7 | 37.9 | 51.2 | 48.3 | 48.5 | 58.4 | 63.7 | 71.1 | 70.4 | 82.3 | 88.0 | 88.5 | 90.4 | 60.9 |
| CoTTA [51] | 40.1 | 37.7 | 39.7 | 26.9 | 38.0 | 27.9 | 26.4 | 32.8 | 31.8 | 40.3 | 24.7 | 26.9 | 32.5 | 28.3 | 33.5 | 32.5 |
| RoTTA [56] | 49.1 | 44.9 | 45.5 | 30.2 | 42.7 | 29.5 | 26.1 | 32.2 | 30.7 | 37.5 | 24.7 | 26.9 | 32.5 | 28.3 | 33.5 | 32.5 |
| PETAL [2] | 38.3 | 36.4 | 38.6 | 25.9 | 36.8 | 27.3 | 25.4 | 32.0 | 30.8 | 38.7 | 24.4 | 26.4 | 31.5 | 26.9 | 32.5 | 31.5 |
| SATA [6] | 36.5 | 33.1 | **35.1** | 25.9 | 34.9 | 27.7 | 25.4 | 29.5 | 29.9 | 33.1 | 23.6 | 26.7 | 31.9 | 27.5 | 35.2 | 30.3 |
| DSS [52] | 39.7 | 36.0 | 37.2 | 26.3 | 35.6 | 27.5 | 25.1 | 31.4 | 30.0 | 37.8 | 24.2 | 26.0 | 30.0 | 26.3 | 31.1 | 30.9 |
| SWA [55] | 39.4 | 36.4 | 37.4 | 25.0 | 36.0 | 26.6 | 25.0 | 29.1 | 28.4 | 35.0 | 23.5 | 25.1 | 28.5 | 25.8 | **29.6** | 30.0 |
| VCoTTA (Ours) | **35.3** | **32.8** | 38.9 | **23.8** | **34.6** | **25.5** | **23.2** | **27.5** | **26.7** | **30.4** | **22.1** | **23.0** | **28.1** | **24.2** | 30.4 | **28.4** |

## 5 Experiment

### 5.1 Experimental Setting

**Dataset**. In our experiments, we employ the CIFAR10C, CIFAR100C, and ImageNetC datasets as benchmarks to assess the robustness of classification models. Each dataset comprises 15 distinct types of corruption, each applied at five different levels of severity (from 1 to 5). These corruptions are systematically applied to test images from the original CIFAR10 and CIFAR100 datasets, as well as validation images from the original ImageNet dataset. For simplicity in tables, we use C1 to C15 to represent the 15 types of corruption, *i.e.*, C1: Gaussian, C2: Shot, C3: Impulse C4: Defocus, C5: Glass, C6: Motion, C7: Zoom, C8: Snow, C9: Frost, C10: Fog, C11: Brightness, C12: Contrast, C13: Elastic, C14: Pixelate, C15: Jpeg.

**Pretrained Model**. Following previous studies [50, 51], we adopt pretrained WideResNet-28 [57] model for CIFAR10to-CIFAR10C, pretrained ResNeXt-29 [54] for CIFAR100-to-CIFAR100C, and standard pretrained ResNet-50 [21] for ImageNet-to-ImagenetC. Note in our VCoTTA [51], we further warm up the pretrained model to obtain the stochastic dynamics for each dataset. Similar to CoTTA, we update all the trainable parameters in all experiments. The augmentation number is set to 32 for all compared methods that use the augmentation strategy.

### 5.2 Methods to be Compared

We compare our VCoTTA with multiple state-of-the-art (SOTA) methods. SOURCE denotes the baseline pretrained model without any adaptation. BN [30, 43] keeps the network parameters frozen, but only updates Batch Normalization. TENT [50] updates via Shannon entropy for unlabeled test data. CoTTA [51] builds the MT structure and uses randomly restoring parameters to the source model. SATA [6] modifies the batch-norm affine parameters using source anchoring-based self-distillation to ensure the model incorporates knowledge of newly encountered domains while avoiding catastrophic forgetting. SWA [55] refines the pseudo-label learning process from the perspective of the instantaneous and long-term impact of noisy pseudo-labels. PETAL [2] tries to estimate the uncertainty in CTTA, which is similar to BNN, but it ignores the unreliable prior problem. All compared methods adopt the same backbone, pretrained model and hyperparameters.

Table 3: Classification error rate (%) for the standard ImageNet-to-ImageNetC CTTA task.

| Method | C1 | C2 | C3 | C4 | C5 | C6 | C7 | C8 | C9 | C10 | C11 | C12 | C13 | C14 | C15 | Avg |
|---|---|---|---|---|---|---|---|---|---|---|---|---|---|---|---|---|
| Source | 95.3 | 95.0 | 95.3 | 86.1 | 91.9 | 87.4 | 77.9 | 85.1 | 79.9 | 79.0 | 45.4 | 96.2 | 86.6 | 77.5 | 66.1 | 83.0 |
| BN | 87.7 | 87.4 | 87.8 | 88.0 | 87.7 | 78.3 | 63.9 | 67.4 | 70.3 | 54.7 | 36.4 | 88.7 | 58.0 | 56.6 | 67.0 | 72.0 |
| Tent [50] | 85.6 | 79.9 | **78.3** | **82.0** | **79.5** | 71.4 | 59.5 | 65.8 | 66.4 | 55.2 | 40.4 | 80.4 | 55.6 | 53.5 | 59.3 | 67.5 |
| CoTTA [51] | 87.4 | 86.0 | 84.5 | 85.9 | 83.9 | 74.3 | 62.6 | 63.2 | 63.6 | 51.9 | 38.4 | 72.7 | 50.4 | 45.4 | 50.2 | 66.7 |
| RoTTA [56] | 88.3 | 82.8 | 82.1 | 91.3 | 83.7 | 72.9 | 59.4 | 66.2 | 64.3 | 53.3 | **35.6** | 74.5 | 54.3 | 48.2 | 52.6 | 67.3 |
| PETAL [2] | 87.4 | 85.8 | 84.4 | 85.0 | 83.9 | 74.4 | 63.1 | 63.5 | 64.0 | 52.4 | 40.0 | 74.0 | 51.7 | 45.2 | 51.0 | 67.1 |
| DSS [52] | 84.6 | 80.4 | 78.7 | 83.9 | 79.8 | 74.9 | 62.9 | 62.8 | 62.9 | 49.7 | 37.4 | **71.0** | 49.5 | 42.9 | 48.2 | 64.6 |
| VCoTTA (Ours) | **81.8** | **78.9** | 80.0 | 83.4 | 81.4 | **70.8** | **60.3** | **61.1** | **61.7** | **46.4** | 35.7 | 71.7 | 50.1 | 47.1 | 52.9 | **64.2** |

## 5.3 Comparison Results

We show the major comparisons with the SOTA methods in *Tables* 1, 2 and 3. We have the following observations. First, no adaptation at the test time (SOURCE) suffers from serious domain shift, which shows the necessity of the CTTA. Second, traditional TTA methods that ignore the continual shift in test time perform poorly such as TENT and BN. We also find that simple Shannon entropy is effective in the first several domain shifts, especially in complex 1,000-classes ImageNetC, but shows significant performance drops in the following shifts. Third, the mean-teacher structure is very useful in CTTA, such as CoTTA and PETAL, which means that the pseudo-label is useful in domain shift. In the previous method, the error accumulation leads to the unreliable pseudo labels, then the model may get more negative transfers in CTTA along the timeline. The proposed VCoTTA outperforms other methods on all the three datasets, such as 13.1% vs. 15.3% (SWA) on CIFAR10C, 28.4% vs. 30.0% (SWA) on CIFAR100C and 64.2% vs. 66.7% (CoTTA) on ImageNetC. We hold the opinion that the prior will inevitably drift in CTTA, but VCoTTA slows down the process via the prior mixture. We also find that the superiority is more obvious in the early adaptation, which may be influenced by the different corruption orders. We analyze the order problem in Appendix H.

## 5.4 Ablation Study

We evaluate the two components in Table 4, *i.e.*, the Variational Warm-Up (VWU) and the Symmetric Cross-Entropy (SCE) via ablation. The ablation results show that the two components are both important for VCoTTA. First, the VWU is used to inject stochastic dynamics into an off-the-shelf pretrained model. Without the VWU, the performance of VCoTTA drops to 18.4% from 13.9% on CIFAR10C, 31.5% from 28.8% on CIFAR100C and 68.1% from 64.2% on ImageNetC. Also, the SCE can further improve the performance on CIFAR10C and CIFAR100C, because SCE balances the gradient for high and low confidence predictions. We also find that SCE is ineffective for complex ImageNetC, and the reason may be the class sensitivity imbalance, causing the model to lean more towards one direction during optimization.

Table 4: Ablation study on under severity 5.

| No. | VWU | SCE | CIFAR10C | CIFAR100C | ImageNetC |
|---|---|---|---|---|---|
| 1 | | | 18.4 | 31.5 | 68.1 |
| 2 | | √ | 17.1 | 31.2 | 68.3 |
| 3 | √ | | 13.9 | 28.8 | **64.2** |
| 4 | √ | √ | **13.1** | **28.4** | 64.7 |

Table 5: Different weights for mixture of priors.

| No. | $\alpha$ | $1-\alpha$ | CIFAR10C | CIFAR100C | ImageNetC |
|---|---|---|---|---|---|
| 1 | 1 | 0 | 17.4 | 35.0 | 69.9 |
| 2 | 0 | 1 | 16.3 | 33.7 | 71.2 |
| 3 | 0.5 | 0.5 | 14.7 | 31.3 | 67.0 |
| 4 | Eq. (10) | | **13.1** | **28.4** | **64.7** |

## 5.5 Mixture of Priors

In Sec. 4.2, we introduce a Gaussian mixture strategy, where the current prior is approximated as the weighted sum of the source prior and the teacher prior. The weights are determined by computing the entropy over multiple augmentations of two models. To assess the effectiveness of these weights, we compare them with three naive weighting configurations: using only the source model, using only the teacher model, and a simple average with equal weights for both models. The results, as presented in Table 5, reveal that relying solely on the source model or the teacher model (i.e., weighting with $(1, 0)$ and $(0, 1)$) results in suboptimal performance. Additionally, naive weighting with equal contributions from both models (i.e., $(0.5, 0.5)$) proves ineffective for CTTA due to the inherent uncertainty in both models. In contrast, the proposed adaptive weights for the Gaussian mixture in CTTA demonstrate its effectiveness. This underscores the significance of striking a balance between the two prior models in

299 an unsupervised environment. The trade-off implies the need to discern when the source model's
300 knowledge is more applicable and when the teacher model's shifting knowledge takes precedence.

## 5.6 Uncertainty Estimation

302 To evaluate the uncertainty estimation, we use negative loglikelihood (NLL) and Brier Score (BS) [3].
303 Both NLL and BS are proper scoring rules [19], and they are minimized if and only if the predicted
304 distribution becomes identical to the actual distribution:

$$\text{NLL} = -\mathbb{E}_{(x,y)\in\mathcal{D}^{\text{test}}} \log(p(y|x,\boldsymbol{\theta})), \quad \text{BS} = \mathbb{E}_{(x,y)\in\mathcal{D}^{\text{test}}} \left(p(y|x,\boldsymbol{\theta}) - \text{Onehot}(y)\right)^2,$$

305 where $\mathcal{D}^{\text{test}}$ denotes the test set, *i.e.*, the unsupervised test dataset $\mathcal{U}$ with labels. We evaluate NLL and
306 BS with a severity level of 5 for all corruption types, and the compared results with SOTAs are shown
307 in Table 6. We have the following observations. First, most methods suffer from low confidence in
308 terms of NLL and BS because of the drift priors, where the model is unreliable gradually, and the error
309 accumulation makes the model perform poorly. Our approach outperforms most other approaches in
310 terms of NLL and BS, demonstrating the superiority in improving uncertainty estimation. We also
311 find that PETAL [2] shows good NLL and BS, because PETAL forces the prediction over-confident
312 to unreliable priors, thus PETAL shows unsatisfactory results on adaptation accuracy, such as 31.5%
313 vs. 28.4% (Ours) on CIFAR100C.

Table 6: Uncertainty estimation via NLL and BS.

| Method | CIFAR10C | | CIFAR100C | | ImageNetC | |
|---|---|---|---|---|---|---|
| | NLL | BS | NLL | BS | NLL | BS |
| Source | 3.0566 | 0.7478 | 2.4933 | 0.6707 | 5.0703 | 0.9460 |
| BN | 0.9988 | 0.3354 | 1.3932 | 0.4740 | 3.9971 | 0.8345 |
| Tent | 1.9391 | 0.3713 | 7.1097 | 1.0838 | 3.6902 | 0.8281 |
| CoTTA | 0.7192 | 0.2761 | 1.2907 | 0.4433 | 3.6235 | **0.7972** |
| PETAL | 0.5899 | 0.2458 | **1.2267** | 0.4327 | 3.6391 | 0.8017 |
| VCoTTA | **0.5421** | **0.2130** | 1.2287 | **0.4307** | **3.4469** | 0.8092 |

Table 7: Gradually changing on severity 5.

| Method | CIFAR10C | CIFAR100C | ImageNetC |
|---|---|---|---|
| Source | 23.9 | 32.9 | 81.7 |
| BN | 13.5 | 29.7 | 54.1 |
| TENT | 39.1 | 72.7 | 53.7 |
| CoTTA | 10.6 | 26.3 | 42.1 |
| PETAL | 10.5 | 27.1 | 60.5 |
| VCoTTA | **8.9** | **24.4** | **39.9** |

## 5.7 Gradually Corruption

315 We also show gradual corruption results instead of constant severity in the major comparison, and the
316 results are reported in Table 7. Specifically, each corruption adopts the gradual changing sequence:
317 $1 \rightarrow 2 \rightarrow 3 \rightarrow 4 \rightarrow 5 \rightarrow 4 \rightarrow 3 \rightarrow 2 \rightarrow 1$, where the severity level is the lowest 1 when corruption
318 type changes, therefore, the type change is gradual. The distribution shift within each type is also
319 gradual. Under this situation, our VCoTTA also outperforms other methods, such as 8.9% vs. 10.5%
320 (PETAL) on CIFAR10C, and 24.4% vs. 26.3% (CoTTA) on CIFAR100C. The results show that the
321 proposed VCoTTA based on BNN is also effective when the distribution change is uncertain.

# 6 Conclusion and Limitation

323 **Conclusion**: In this paper, we proposed a variational Bayesian inference approach, termed VCoTTA,
324 to estimate uncertainties in CTTA. At the pretrained stage, we first transformed an off-the-shelf
325 pretrained deterministic CNN into a BNN using a variational warm-up strategy, thereby injecting
326 uncertainty into the source model. At the test time, we implemented a mean-teacher update strategy,
327 where the student model is updated via variational inference, while the teacher model is refined by the
328 exponential moving average. Specifically, to update the student model, we proposed a novel approach
329 that utilizes a mixture of priors from both the source and teacher models. Consequently, the ELBO
330 can be formulated as the cross-entropy between the student and teacher models, combined with the
331 KL divergence of the prior mixture. We demonstrated the effectiveness of the proposed method on
332 three datasets, and the results show that the proposed method can mitigate the issue of unreliable
333 prior within the CTTA framework.

334 **Limitation**: The efficacy of the proposed method relies on injecting uncertainty into the model during
335 the pre-training phase, which may be unavailable in scenarios where pretraining is already completed,
336 and original data is inaccessible. Additionally, constructing and training BNN models are inherently
337 more complex compared to CNNs, highlighting the importance of enhancing computational efficiency.
338 The Gaussian mixture method relies on multiple data augmentations, which also incurs computational
339 costs. Future endeavors could explore more efficient approaches for Gaussian mixture.

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

# Variational Continual Test-Time Adaptation
## (Appendix)

## A Bayesian Inference (BI) in Traditional CL and CTTA

As described in Sec. 3, we first illustrate the BI has been studied in traditional Continual Learning (CL) methods. In this section, we compare the BI in CL and CTTA in detail and show the differences with some related works. The comparison can be seen in Fig. 3. For the CL, BI is conducted by the posterior propagation, that is, the prior of next task is equal to the current posterior. This is feasible in supervised CL, where the data label is provided. For the CTTA, the posterior is not trustworthy using only pseudo labels to adapt to a new domain. Thus, propagate the untrustworthy posterior to the next stage would make unreliable prior, which will result in error accumulation. In the proposed VCoTTA, we propose to solve the problem via enhancing the two terms in VI (see Sec. 4).

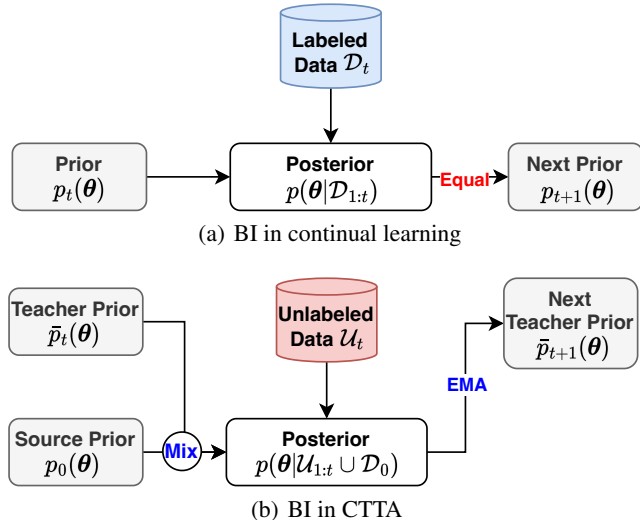

(a) BI in continual learning

(b) BI in CTTA

Figure 3: Bayesian inference comparison between continual learning and CTTA. We find the traditional prior transmission is infeasible in CTTA because of the unreliable prior from unlabeled data. In our method, we place CTTA in a mean-teacher structure, and design BI in CTTA using a mixture of teacher prior and source prior. The next teacher prior is updated by the exponential moving average.

VCL [38] is a classic CL study that uses VI, our work is also inspired by VCL but has the following difference. (1) *The tasks are different*: VCL studies supervised CL task, while our VCoTTA studies unsupervised CTTA task. (2) *The challenges are differnt*: CL only suffers from catastrophic forgetting (CF), while CTTA sufffers from both CF and error accumulation. (3) *Ways of BI are different*: To conduct BI, one needs to compute prior and likelihood. For the prior, the current prior of VCL is set to be the previous posterior, while in CTTA such a prior may be unreliable. For the likelihood, VCL can directly compute likelihood, CTTA is under unsupervised setting, thus in our work, we deduce the BI in CTTA using conditional entropy. (4) *The update strategies are different*: To reduce error accumulation in unsupervised scenario, we employ a mean-teacher update strategy using VI for the student model and exponential moving average for the teacher model, and compute a prior mixture to guide the student update. Moreover, VCL maintains an extra coreset from the training set, while VCoTTA never store any data during the test time.

We also find another recent work named PETAL [2] that estimates uncertainties in CTTA. The BI formulation is similar between PETAL and ours, which is derived from [20], but PETAL use different method to conduct the inference: (1) PETAL only uses CNN and does not estimate the *model* uncertainties, while VCoTTA uses BNN to model the uncertainties during test time. (2) PETAL ignores the unreliable prior in CTTA, and follow the VCL setting that use the previous posterior as the current prior. (3) We conduct BI using variational inference while PETAL use SWAG [35]. SWAG has advantages in terms of computational efficiency and stability during training, especially in scenarios where computational resources are limited. However, SWAG might not handle unreliable priors as effectively as VI since it doesn't explicitly model the posterior distribution. (4) We have compared with PETAL in our experiment (see Tables 1, 2, 3), and our method outperforms PETAL on all datasets.

# B    CTTA Approximation by BI

## B.1    Assumption on Class Separability

In our method, we use the conditional entropy to alternate the intractable computing of likelihood. Note that the use of entropy in unsupervised scenario needs to satisfy the class-separable assumption. In fact, unlabeled data do not convey category information but still carry information. Miller and Uyar [36] theoretically proved that utilizing unlabeled samples to train classifiers can improve classification performance if there is a connection between the target and sample distributions. It is a common practice in unsupervised/semi-supervised learning to establish the relationship between unlabeled data and the target by making some reasonable assumptions to obtain category-relevant information from unlabeled data. Common assumptions include the *Smoothness assumption*, *Cluster assumption*, *Manifold assumption*, *Low-density separation assumption*, etc. For example, the well-known clustering-based methods utilize the cluster assumption to generate pseudo-labels for unsupervised learning [48]. Caron et al. [5] assumes that "the model trained on labeled data will produce high uncertainty estimation for unseen data" in domain adaptation tasks to benefit the classifier from unlabeled data lacking category information.

Bengio et al. in [20] proposed the conditional entropy and point out that "These studies conclude that the (asymptotic) information content of unlabeled examples decreases as classes overlap. Thus, the assumption that classes are well separated is *sensible* if we expect to take advantage of unlabeled examples." This assumption has been applied to many studies, for example in [29, 33, 60, 2]. In the CTTA task of this paper, as the task progresses, the domain shifts, but the categories in the task remain unchanged. Therefore, under the assumption that unlabeled data contains information, we can reasonably continue to use conditional entropy in the current scenario. To sum up, whether in unsupervised TTA or in the Bayesian field, this assumption is not difficult to achieve or has never been applied. We can quite naturally continue to use this assumption in the context of this paper.

## B.2    BI during Test Time

The goal of CTTA is to learn a posterior distribution $p(\boldsymbol{\theta}|\mathcal{U}_{1:T} \cup \mathcal{D}_0)$ from a source dataset $\mathcal{D}_0$, and a sequence of unlabeled test data from $\mathcal{U}_1$ to $\mathcal{U}_T$. Following [60], assuming we have multiple input-generating distributions that the source dataset $\mathcal{D}_0$ is drawn from a distribution $\phi$, and $\tilde{\phi}_t$ specifies the shifted of the $t$-th unlabeled test dataset which we aim to adapt to. Let the parameters of the model be $\boldsymbol{\theta}$,then following the semi-supervised learning framework [20], we incorporate all input-generating distributions into the belief over the model parameters $\boldsymbol{\theta}$ as follows

$$p(\boldsymbol{\theta}|\phi, \tilde{\phi}_1, \cdots, \tilde{\phi}_T) \propto p(\boldsymbol{\theta}) \exp\left(-\lambda_0 H_{\boldsymbol{\theta}, \phi}(Y|\boldsymbol{X})\right) \prod_{t=1}^{T} \exp\left(-\lambda_t H_{\boldsymbol{\theta}, \tilde{\phi}_t}(Y|\boldsymbol{X})\right), \qquad (18)$$

where the inputs $X$ are sampled i.i.d. from a generative model with parameters $\phi$, while the corresponding labels $Y$ are sampled from a conditional distribution $p(Y|X, \boldsymbol{\theta})$, which is parameterized by the model parameters $\boldsymbol{\theta}$. $p(\boldsymbol{\theta})$ is a prior distribution over $\boldsymbol{\theta}$. $\{\lambda_0, \lambda_1, \cdots, \lambda_T\}$ are the factors for approximation weighting. Generally, the entropy term $H_{\boldsymbol{\theta}, \phi}(Y|\boldsymbol{X})$ represents the cross entropy of the supervised learning, and the entropy term $H_{\boldsymbol{\theta}, \tilde{\phi}_t}(Y|\boldsymbol{X})$ for $t > 0$ denotes the Shannon entropy of the unsupervised learning.

554 Following [60], we can empirically use a point estimation to get a plug-in Bayesian approach to
555 approximate the above formula:

$$p(\boldsymbol{\theta}|\mathcal{U}_{1:T} \cup \mathcal{D}_0)$$

$$\propto \quad p(\boldsymbol{\theta}) \prod_{\forall x,y \in \mathcal{D}_0} p(y|x,\boldsymbol{\theta}) \exp\left(-\frac{\lambda_0}{|\mathcal{D}_0|}\sum_{\forall x \in \mathcal{D}_0} H(Y|x,\boldsymbol{\theta})\right) \prod_{t=1}^{T} \exp\left(-\frac{\lambda_t}{|\mathcal{U}_t|}\sum_{\forall x \in \mathcal{U}_t} H(Y|x,\boldsymbol{\theta})\right). \tag{19}$$

556 To make the formula feasible to CTTA, that is, no source data is available at the test time, we set
557 $\lambda_0 = 0$. And the source knowledge can be represented by $p(\boldsymbol{\theta}|\mathcal{D}_0) \propto p(\boldsymbol{\theta}) \prod_{\forall x,y \in \mathcal{D}_0} p(y|x,\boldsymbol{\theta})$.
558 Thus, for the $t$-th test domain, the Bayesian inference in CTTA can be represented as follows:

$$p(\boldsymbol{\theta}|\mathcal{U}_{1:t} \cup \mathcal{D}_0) \propto p(\boldsymbol{\theta}|\mathcal{D}_0) \prod_{i=1}^{t} \exp\left(-\frac{\lambda_i}{|\mathcal{U}_i|}\sum_{\forall x \in \mathcal{U}_i} H(Y|x,\boldsymbol{\theta})\right)$$

$$\propto p(\boldsymbol{\theta}|\mathcal{U}_{1:t-1} \cup \mathcal{D}_0) \exp\left(-\frac{\lambda_t}{|\mathcal{U}_t|}\sum_{\forall x \in \mathcal{U}_t} H(Y|x,\boldsymbol{\theta})\right), \tag{20}$$

559 where $H(\mathcal{U}_t|\boldsymbol{\theta}) = \frac{1}{|\mathcal{U}_t|}\sum_{\forall x \in \mathcal{U}_t} H(Y|x,\boldsymbol{\theta})$ and the above formula can be rewritten in simplicity as

$$p(\boldsymbol{\theta}|\mathcal{U}_{1:t} \cup \mathcal{D}_0) \propto p(\boldsymbol{\theta}|\mathcal{U}_{1:t-1} \cup \mathcal{D}_0)e^{-\lambda H(\mathcal{U}_t|\boldsymbol{\theta})} = p_t(\boldsymbol{\theta})e^{-\lambda H(\mathcal{U}_t|\boldsymbol{\theta})}, \tag{21}$$

560 which specifies the Bayesian inference process on continuously arriving unlabeled data in CTTA.

## 561 C ELBO of the VI in CTTA

562 We built VI for CTTA in Sec. 3, where we initialize a variational distribution $q(\boldsymbol{\theta})$ to approximate the
563 real posterior. For the test domain $t$, we optimize the variational distribution as follows:

$$q_t(\boldsymbol{\theta}) = \arg\min_{q \in \mathbb{Q}} \mathrm{KL}\left[q(\boldsymbol{\theta}) \,\|\, \frac{1}{Z_t}p_t(\boldsymbol{\theta})e^{-\lambda H(\mathcal{U}_t|\boldsymbol{\theta})}\right], \tag{22}$$

564 where $\mathbb{Q}$ is the distribution searching space, and $p_t(\boldsymbol{\theta})$ is the current prior.

565 Following the definition of KL divergence and the standard derivation of the Evidence Lower BOund
566 (ELBO) is as the following formulas. Specifically, the KL divergence is expanded as

$$\mathrm{KL}\left[q(\boldsymbol{\theta}) \,\|\, \frac{1}{Z_t}p_t(\boldsymbol{\theta})e^{-\lambda H(\mathcal{U}_t|\boldsymbol{\theta})}\right]$$

$$= -\int_{\boldsymbol{\theta}} q(\boldsymbol{\theta}) \log \frac{\frac{1}{Z_t}p_t(\boldsymbol{\theta})e^{-\lambda H(\mathcal{U}_t|\boldsymbol{\theta})}}{q(\boldsymbol{\theta})} d\boldsymbol{\theta}$$

$$= -\int_{\boldsymbol{\theta}} q(\boldsymbol{\theta}) \log \frac{1}{Z_t}e^{-\lambda H(\mathcal{U}_t|\boldsymbol{\theta})} d\boldsymbol{\theta} - \int_{\boldsymbol{\theta}} q(\boldsymbol{\theta}) \log \frac{p_t(\boldsymbol{\theta})}{q(\boldsymbol{\theta})} d\boldsymbol{\theta} \tag{23}$$

$$= \int_{\boldsymbol{\theta}} q(\boldsymbol{\theta}) \log Z_t d\boldsymbol{\theta} + \lambda \int_{\boldsymbol{\theta}} q(\boldsymbol{\theta})H(\mathcal{U}_t|\boldsymbol{\theta}) d\boldsymbol{\theta} - \int_{\boldsymbol{\theta}} q(\boldsymbol{\theta}) \log \frac{p_t(\boldsymbol{\theta})}{q(\boldsymbol{\theta})} d\boldsymbol{\theta}$$

$$= \log Z_t + \lambda \mathbb{E}_{\boldsymbol{\theta} \sim q(\boldsymbol{\theta})} H(\mathcal{U}_t|\boldsymbol{\theta}) + \mathrm{KL}\left(q(\boldsymbol{\theta}) \,\|\, p_t(\boldsymbol{\theta})\right),$$

567 where the first constant term can be reduced in the optimization. Thus, we can optimize the variational
568 distribution via the ELBO:

$$q_t(\boldsymbol{\theta}) = \arg\min_{q \in \mathbb{Q}} \mathrm{KL}\left[q(\boldsymbol{\theta}) \,\|\, \frac{1}{Z_t}p_t(\boldsymbol{\theta})e^{-\lambda H(\mathcal{U}_t|\boldsymbol{\theta})}\right]$$

$$= \arg\max_{q \in \mathbb{Q}} -\lambda \mathbb{E}_{\boldsymbol{\theta} \sim q(\boldsymbol{\theta})} H(\mathcal{U}_t|\boldsymbol{\theta}) - \mathrm{KL}\left(q(\boldsymbol{\theta}) \,\|\, p_t(\boldsymbol{\theta})\right) \tag{24}$$

$$= \arg\max_{q \in \mathbb{Q}} \mathrm{ELBO}.$$

In our case, the former entropy term can be more effectively replaced by the cross entropy or symmetric cross entropy (SCE) between the student model and the teacher model in a mean-teacher architecture (see Sec. 4.1). For the latter KL term, we can substitute a variational approximation that we deem closest to the current-stage prior $p_t(\boldsymbol{\theta})$ into the KL divergence. When the prior is a multivariate Gaussian distribution, this term can be computed in closed form as

$$
\begin{aligned}
&\mathrm{KL}\left(\mathcal{N}(\boldsymbol{\mu}_1, \boldsymbol{\Sigma}_1) \parallel \mathcal{N}(\boldsymbol{\mu}_2, \boldsymbol{\Sigma}_2)\right) \\
&= \frac{1}{2}\left(\mathrm{tr}(\boldsymbol{\Sigma}_2^{-1}\boldsymbol{\Sigma}_1) + (\boldsymbol{\mu}_2 - \boldsymbol{\mu}_1)^{\top}\boldsymbol{\Sigma}_2^{-1}(\boldsymbol{\mu}_2 - \boldsymbol{\mu}_1) - k + \ln\left(\frac{\det(\boldsymbol{\Sigma}_2)}{\det(\boldsymbol{\Sigma}_1)}\right)\right).
\end{aligned} \tag{25}
$$

where $\boldsymbol{\Sigma} = \mathrm{diag}(\boldsymbol{\sigma}^2)$, $k$ represents the dimensionality of the distributions, $\mathrm{tr}(\cdot)$ denotes the trace of a matrix, and $\det(\cdot)$ stands for the determinant of a matrix. For the case that the prior is a mixture of Gaussian distributions, we can refer to the next section to get its upper bound.

# D  Mixture-of-Gaussian Prior

## D.1  Upper Bound of the Mixture of Two KL Divergencies

We refer to the lemma that was stated for the mixture of Gaussian in [44]. The KL divergence between two mixture distributions $p = \sum_{i=1}^{k} \alpha_i p_i$ and $p' = \sum_{i=1}^{k} \alpha_i p'_i$ is upper-bounded by

$$
\mathrm{KL}(p \parallel p') \le \mathrm{KL}(\boldsymbol{\alpha} \parallel \boldsymbol{\alpha}') + \sum_{i=1}^{k} \alpha_i \mathrm{KL}(p_i \parallel p'_i), \tag{26}
$$

where $\boldsymbol{\alpha} = (\alpha_1, \alpha_2, \cdots, \alpha_k)$ and $\boldsymbol{\alpha}' = (\alpha'_1, \alpha'_2, \cdots, \alpha'_k)$ are the weights of the mixture components. The equality holds if and only if $\alpha_i p_i / \sum_{j=1}^{k} \alpha_j p_j = \alpha'_i p'_i / \sum_{j=1}^{k} \alpha'_j p'_j$ for all $i$. Using the log-sum inequality [10], we have

$$
\begin{aligned}
\mathrm{KL}(\sum_{i=1}^{k} \alpha_i p_i \parallel \sum_{i=1}^{k} \alpha_i p'_i) &= \int \left(\sum_{i=1}^{k} \alpha_i p_i\right) \log \frac{\sum_{i=1}^{k} \alpha_i p_i}{\sum_{i=1}^{k} \alpha_i p'_i} \\
&\le \int \sum_{i=1}^{k} \alpha_i p_i \log \frac{\alpha_i p_i}{\alpha_i p'_i} \\
&= \sum_{i=1}^{k} \alpha_i \left(\int p_i \log \frac{\alpha_i}{\alpha'_i} + \int p_i \log \frac{p_i}{p'_i}\right) \\
&= \mathrm{KL}(\boldsymbol{\alpha} \parallel \boldsymbol{\alpha}') + \sum_{i=1}^{k} \alpha_i \mathrm{KL}(p_i \parallel p'_i).
\end{aligned}
$$

In our algorithm, $q(\boldsymbol{\theta})$ is set to be a mixture of Gaussian distributions, *i.e.*, $p_t(\boldsymbol{\theta}) = \alpha \cdot p_1(\boldsymbol{\theta}) + (1 - \alpha) \cdot \bar{p}_t(\boldsymbol{\theta})$. In the above inequality, let $q(\boldsymbol{\theta}) = \sum_{i=1}^{k} \alpha_i q(\boldsymbol{\theta})$, we can get the upper bound of the KL divergence between $q(\boldsymbol{\theta})$ and $p_t(\boldsymbol{\theta})$:

$$
\mathrm{KL}(q \parallel p_t) \le \alpha \cdot \mathrm{KL}(q \| p_1) + (1 - \alpha) \cdot \mathrm{KL}(q \| \bar{p}_t). \tag{27}
$$

So the lower bound (24) can be redefined as

$$
\begin{aligned}
\mathcal{L} &= -\lambda \mathbb{E}_{\boldsymbol{\theta} \sim q(\boldsymbol{\theta})} H(\mathcal{U}_t | \boldsymbol{\theta}) - \mathrm{KL}\left(q(\boldsymbol{\theta}) \parallel p_t(\boldsymbol{\theta})\right) \\
&\ge -\lambda \mathbb{E}_{\boldsymbol{\theta} \sim q(\boldsymbol{\theta})} H(\mathcal{U}_t | \boldsymbol{\theta}) - \alpha \cdot \mathrm{KL}(q \| p_1) - (1 - \alpha) \cdot \mathrm{KL}(q \| \bar{p}_t) \\
&\stackrel{\mathrm{def}}{=} \mathcal{L}',
\end{aligned} \tag{28}
$$

Then, we have obtained a lower bound that can be optimized through closed-form calculations as the source prior distribution $q_0(\boldsymbol{\theta})$ and the teacher prior distribution $\bar{q}_t(\boldsymbol{\theta})$ are multivariate Gaussian distributions, which means we can also optimize $\mathcal{L}'$ with Eq. (25).

## D.2 Advantage of the Mixture of Gaussian Prior

In this subsection, we illustrate why the mixture of Gaussian prior are beneficial to CTTA. First of all, we can start from defining what is a better distribution for CTTA. Assume there exists an ideal prior distribution $\hat{p}_t$, which effectively represents the distribution of the model after learning all past knowledge, including that from the source and unlabeled datasets. Then we can use the difference between a distribution and the ideal distribution $\hat{p}_t$ (here we use KL divergence) to measure the goodness of a distribution, i.e., $\mathrm{KL}(\cdot||\hat{p}_t)$.

Generally, neither the source prior $p_1$ (trained on labeled data) nor the adapted prior $\bar{p}_t$ (adapt on unlabeled data, being unreliable) can be completely consistent with $\hat{p}_t$. Considering that, as $t$ increases, the difference between $\bar{p}_t$ and $\hat{p}_t$ will increase without an upper bound due to the error accumulation (since $t$ is infinitely growing). The source prior $p_1$ cannot adapt to the unlabeled data, but it contains important information from the labeled data, and the ideal distribution cannot forget the source information too much, so we can assume that the difference between $p_1$ and $\hat{p}_t$ is a constant, i.e., $\mathrm{KL}(p_1||\hat{p}_t) < U$, where $U$ is a constant upper bound. Accordingly, it can be considered that mixing the source prior $p_1$ and the adapted prior $\bar{p}_t$ in some way is beneficial for reducing $\mathrm{KL}(\cdot||\hat{p}_t)$.

In our paper, we consider using a simple Gaussian mixture, i.e., $p_t = \alpha_t p_1 + (1 - \alpha_t)\bar{p}_t$, where $\alpha$ is computed by Eq. (10). It is easy to illustrate the benefits of this idea using the following inequality:

$$\begin{aligned}
\mathrm{KL}(p_t||\hat{p}_t) &= \mathrm{KL}\left[(\alpha_t p_1 + (1 - \alpha_t)\bar{p}_t)||\hat{p}_t\right] \\
&\leq \alpha_t \mathrm{KL}(p_1||\hat{p}_t) + (1 - \alpha_t)\mathrm{KL}(\bar{p}_t||\hat{p}_t) \\
&\leq \alpha_t U + (1 - \alpha_t)\mathrm{KL}(\bar{p}_t||\hat{p}_t).
\end{aligned} \tag{29}$$

In Eq. (29), if $\mathrm{KL}(\bar{p}_t||\hat{p}_t) \geq U$, which can be satisfied as mentioned above, then we have

$$\mathrm{KL}(p_t||\hat{p}_t) \leq \mathrm{KL}(\bar{p}_t||\hat{p}_t),$$

This indicates that the mixed distribution $p_t$ is closer to the ideal distribution $\hat{p}_t$ than the adapted prior $\bar{p}_t$. A similar idea can be found in the stochatic restoration in CoTTA [51], where the author randomly restore parts of parameters of the current model into the parameters of source model.

# E Augmentation Analysis

In our method, we use the standard augmentation following CoTTA [51]. In this subsection, we analyze the some characteristics via experiments.

## E.1 Confidence Margin

First, we analyze the margin $\epsilon$ in Eq. (13). We experimentally validate different margins with more choices. Experimental results are shown in Tables 8. The results indicate that different datasets may require different margins to control confidence. Moreover, Eq. (13) signifies that the reliable teacher likelihood is represented by the mean of its augmentations with $\epsilon$ more confidence than the teacher itself. Tables 8 illustrates the selection of $\epsilon$ in our approach on CIFAR10C, CIFAR100C and ImageNetC. Note that when $\epsilon = -1$, it means no margin is used and the method will use all augmented samples, i.e., without using Eq. (13). The results show that the proposed margin can effectively filter out unreliable augmented samples and achieve a better teacher log-likelihood.

Table 8: Analysis on confidence margin.

| No. | $\epsilon$ | CIFAR10C | $\epsilon$ | CIFAR100C | $\epsilon$ | ImageNetC |
|-----|------------|----------|------------|-----------|------------|-----------|
| 1 | -1 | 15.1 | -1 | 29.3 | -1 | 66.4 |
| 2 | 0 | 13.23 | 0 | 28.78 | 0 | 65.0 |
| 3 | 1e-4 | 13.23 | 0.1 | 28.55 | 1e-3 | 65.0 |
| 4 | 1e-3 | 13.22 | 0.2 | 28.45 | 1e-2 | 64.8 |
| 5 | 1e-2 | **13.14** | 0.3 | **28.43** | 1e-1 | **64.7** |
| 6 | 1e-1 | 13.31 | 0.4 | 28.54 | 2e-1 | 66.2 |

## E.2 Different Number of Augmentation

In our method, we also use augmentation to enhance the confidence. We then evaluate the the number of augmentation in Eq. (10). The results can be seen in Table 9, and shows that increasing the number of augmentations can enhance effectiveness, but this hyperparameter ceases to have a significant impact after reaching 32.

Table 9: Different number of augmentation.

| Method | 0 | 4 | 8 | 16 | 32 | 64 |
|--------|------|------|------|------|------|------|
| CoTTA | 17.5 | 17.0 | 16.6 | 16.5 | 16.3 | 16.2 |
| PETAL | 17.3 | 16.9 | 16.4 | 16.1 | 16.0 | 16.0 |
| VCoTTA | **14.9** | **13.8** | **13.6** | **13.3** | **13.1** | **13.1** |

# F Further Discussion on Variational Warm-up Strategy

We have discussed the Variational Warm-Up (VWU) strategy in Sec. 4.3.1, and explain that the warm-up strategy is a common practice in TTA and CTTA. In this section, we further discuss some attributes of the proposed variational warm-up strategy.

In our method, the VWU strategy is used to turn an off-the-shelf CNN to a pretrained BNN. The advantage of this approach is that pretrained CNNs are readily available (e.g., directly leveraging official models in PyTorch), while pretrained BNNs are challenging to obtain, especially for large-scale datasets. Moreover, training BNNs is more difficult compared to training CNNs. Therefore, constructing BNN pretrained models based on existing CNN pretrained models is a feasible approach. Additionally, we find that such a warm-up strategy requires only a few epochs to achieve satisfactory results. To validate the characteristics of the proposed VWU strategy, we designed the following experiments.

## F.1 Warm-up on CNN vs. Directly Pretraining BNN

First, we conducted experiments to compare the performance of obtaining pretrained BNN models using the warm-up approach versus directly training the source model with BNN. We pretrain the BNN also use VI as describing in Sec. 4.3.1. The results can be seen in Table 10. As we can see, the results are at the same level, for example VI pretraining is with 13.2% error rate while the proposed VWU achieves 13.1% on CIFAR10C. However, if we direct turn a pretrained CNN to a BNN by adding random stochastic parameters, without warm-up strategy, the results drop to 17.1%. This shows that VWU is a feasible strategy to obtain a pretrained BNN.

Table 10: Error comparison between varional warm-up on CNN and directly pretraining BNN.

| Method | CIFAR10C | CIFAR100C | ImagenetC |
|--------|----------|-----------|-----------|
| BNN (Random) → BNN + VI pretraining | 13.2 | 29.0 | 65.5 |
| CNN (Pretrained) → BNN w/o VWU | 17.1 | 31.2 | 68.3 |
| CNN (Pretrained) → BNN w/ VWU | 13.1 | 28.4 | 64.7 |

## F.2 Number of Warm-up Epochs

In our implementation, we employ only a limited number of epochs for variational warm-up, say 5 epochs. This is due to the fact that the pretrained model fits well in CNN, thus requiring minimal adjustments to the mean of BNN. Additionally, the standard deviation (std) is initialized to be small. Consequently, only a small number of iterations are necessary to update the BNN, and the step size is also kept small. Experimentation on the epoch number of variational warm-up reveals that keeping increasing epochs ( $> 5$ ) will diminishes performance, as shown in Fig. 5.

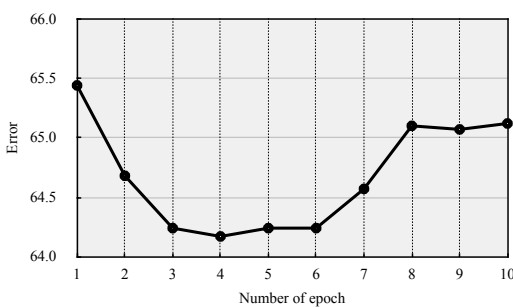
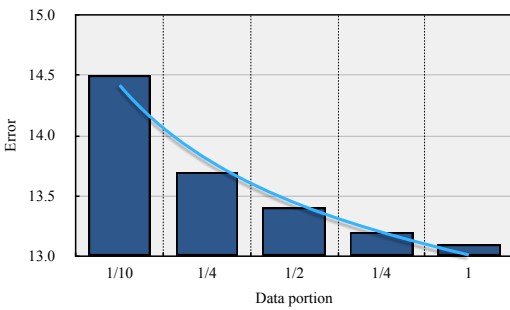

Figure 4: Comparisons on different warm-up epochs (CIFAR10C).

Figure 5: Comparisons on different warm-up data scale (CIFAR10C).

### F.3 Only Portion Usage of Source Dataset in Warm-up

As we response to the weakness, the warm-up strategy is a common approach in TTA and CTTA tasks and it is regarded as a part of pretraining stage. We also evaluate how if we only use partial data for warm-up, and the results are as follow. The experimental results demonstrate that a moderate reduction in sample size still maintains certain effectiveness of the warmup strategy. However, excessive reduction, such as reducing to 1/10, leads to a certain decline in effectiveness. This is because the warmup strategy aims to incorporate statistical information of the dataset into the model, and insufficient data may result in inaccurate performance.

## G Recursive Variational Approximation Process in VCoTTA

In this section, we show the algorithmic workflow utilizing variational approximation in VCoTTA.

**Before testing time**: First, we adopt a variational warm-up strategy to inject stochastic dynamics into the model before adaptation. Given the source dataset $\mathcal{D}_0$, we can use a variational approximation of $p(\boldsymbol{\theta}|\mathcal{D}_0)$ as follows

$$p(\boldsymbol{\theta}|\mathcal{D}_0) = p_1(\boldsymbol{\theta}) \approx q_0(\boldsymbol{\theta}) = \arg\min_{q\in\mathbb{Q}} \mathrm{KL}\left[q(\boldsymbol{\theta}) \parallel \frac{1}{Z_0}p(\boldsymbol{\theta})p(\mathcal{D}_0|\boldsymbol{\theta})\right], \tag{30}$$

where we use the pretrained deterministic model $p_0(\boldsymbol{\theta})$ as the prior distribution.

**When the domain shift**: Then, at the beginning of the test time, we set the prior in task $t$ as $p_t(\boldsymbol{\theta}) = \alpha \cdot p_1(\boldsymbol{\theta}) + (1-\alpha) \cdot \bar{p}_t(\boldsymbol{\theta})$ and variational approximation, where $p_1(\boldsymbol{\theta}) \approx q_0(\boldsymbol{\theta})$ and $\bar{p}_t(\boldsymbol{\theta}) \approx \bar{q}_t(\boldsymbol{\theta})$. For $\bar{q}_t(\boldsymbol{\theta})$, which means the real-time posterior probability of the teacher model for the $t$-th test domain, is constantly updated by $q_t(\boldsymbol{\theta})$ via EMA (see Sec. 4.3.3) during the test phase. Note that we do not have $\bar{q}_t(\boldsymbol{\theta})$ for the first update in the $t$-th phase. In fact, we use $q_{t-1}(\boldsymbol{\theta})$ construct the prior, thus we have $p_t(\boldsymbol{\theta}) \approx \alpha \cdot p_1(\boldsymbol{\theta}) + (1-\alpha) \cdot q_{t-1}(\boldsymbol{\theta})$. This is the variational distribution that should be used to approximate the prior in the absence of a teacher model in the first step, as well as the approximation that should be used when not employing the MT architecture. Note that the process is not required to inform the model that the domain produces a shift.

**During the testing time of a domain**: With the approximation to $p_t(\boldsymbol{\theta})$ and analysis from Appendix B.2, we get $q_t(\boldsymbol{\theta})$ for student model at the test domain $t$ as follows:

$$q_t(\boldsymbol{\theta}) = \arg\min_{q\in\mathbb{Q}} \mathrm{KL}\left[q(\boldsymbol{\theta}) \parallel \frac{1}{Z_t}p_t(\boldsymbol{\theta})e^{-\lambda H(\mathcal{U}_t|\boldsymbol{\theta})}\right], \tag{31}$$

which means, we can recursively derive $p_{t+1}(\boldsymbol{\theta})$ and the following variational distributions, thereby achieving the goal of VCoTTA.

## H Different Orders of Corruption

As we discuss in the major comparisons (see Sec 5.3), the performance may be affected by the corruption order. To provide a more comprehensive evaluation of the matter of the order, we conduct

10 different orders from Sec 5.3, and show the average performance of all compared methods. 10 independent random orders of corruption are all under the severity level of 5. The results are shown in Table 11. We find that the order of corruption is minor on simple datasets such as CIFAR10C and CIFAR100C, but small std on difficult datasets such as ImageNetC. The proposed VCoTTA outperforms other methods on the average error of CIFAR10C and CIFAR100C under 10 different corruption orders, which shows the effectiveness of the prior calibration in CTTA. Moreover, VCoTTA has comparable results with PETAL on ImageNetC, but smaller std over 10 orders, which shows the robustness of the proposed method.

Table 11: Comparisons over 10 orders (avg ± std).

| Method | CIFAR10C | CIFAR100C | ImageNetC |
|--------|----------|-----------|-----------|
| CoTTA | 17.3±0.3 | 32.2±0.3 | 63.4±3.0 |
| PETAL | 16.0±0.1 | 33.8±0.3 | **62.7**±2.6 |
| VCoTTA | **13.1±0.1** | **28.2±0.2** | 62.8±**1.1** |

# I  Corruption Loops

In the real-world scenario, the testing domain may reappear in the future. We evaluate the test conditions continually 10 times to evaluate the long-term adaptation performance on CIFAR10C. That is, the test data will be re-inference and re-adapt for 9 more turns under severity 5. Full results can be found in Fig. 6. The results show that most compared methods obtain performance improvement in the first several loops, but suffer from performance drop in the following loops. This means that the model drift can be even useful in early loops, but the drift becomes hard because of the unreliable prior. The results also indicate that our method outperforms others in this long-term adaptation situation and has only small performance drops.

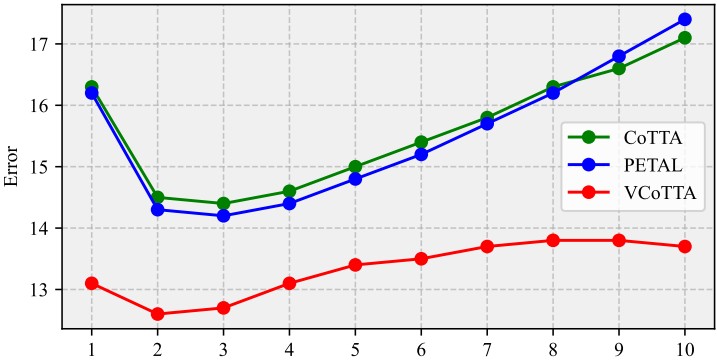

Figure 6: 10 loops under a same corruption order (CIFAR10C).

# J  Experiment on Online Setting

CTTA does operate in an online setting, where all testing data is used only once. However, the current focus of CTTA research primarily revolves around batch-mode online settings, with batch sizes typically set to 200 in our experiments like other SOTAs. In CTTA, strict online learning settings where each data point is processed individually are under-researched. In fact, our method can be applied in scenarios with online learning or small batch sizes. However, it's important to note that the batch normalization (BN) layers is disabled when the batch size is 1. We experimented with batch size of 1 on CIFAR10C, and compare the results with some baseline methods. The comparison results are shown in Table 12. The results show that small batch size in CTTA makes worse performance. We believe this is because a small batch size amplifies the uncertainty in model training.

Table 12: Error comparisons of strict online learning (batch size = 1).

| Method | Batch size 1 | Batch size 200 |
|--------|--------------|----------------|
| TENT   | 43.5         | 20.1           |
| CoTTA  | 42.4         | 16.3           |
| VCoTTA | **39.1**     | **13.1**       |

## K    Time and Memory Cost

We implement our method using a single RTX-4090 GPU card. We provide the memory and time cost in Table 13. Our proposed VCoTTA method does not offer an advantage in terms of memory usage. This is because in the BNN framework, additional standard deviations are required for implementing local reparameterization tricks. However, during the testing phase, this does not significantly impact the efficiency of the model. This is because during testing, only the student model employs variational inference, which requires uncertainty parameters.

Table 13: Time and memory cost comparisons.

| Method | Memory | Time per corruption |
|--------|--------|---------------------|
| CoTTA  | 10.3Gb | 272s                |
| PETAL  | 10.2Gb | 261s                |
| VCoTTA | 11.1Gb | 279s                |

