# OpenReview forum: "Variational Continual Test-Time Adaptation"
_NeurIPS.cc/2024/Conference — Submitted to NeurIPS 2024_

### Official Review · Reviewer_yBx1 · 2024-06-17

**Soundness:** 3
**Presentation:** 2
**Contribution:** 2
**Rating:** 4
**Confidence:** 3

**Summary:**

The authors propose a variational continual adaptation method. Where a sequence of test-time domain adaptation problems are shown to a model. More specifically a labeled dataset is given as an initial dataset to learn from, and then afterwards a sequence of unlabelled datasets with domain shifts are presented to the model.

Using traditional variational continual learning methods will result in error accumulations in the posterior over parameters. So the authors propose a scheme that regularizes against the posterior learned from an initial source dataset.

The authors propose do away with using sequential Bayesian inference. Instead, the authors use a mix of a source prior (learned using labeled data) and a teacher prior. The teacher prior is an EMA of the previous task’s posterior learned with variational inference. The method the authors propose is named VCoTTA.

**Strengths:**

* The paper is well written and the components of VCoTTA are clearly explained.
* The authors provide an ablation to demonstrate which design choices worked well, for instance, to demonstrate that the mixture of priors worked well for test-time adaptation.

**Weaknesses:**

Novelty:
* In terms of novelty I’m not convinced that there are important applications of continual test-time adaptation in the form of classification tasks derived from CIFAR10 datasets. I could be wrong, but some justification in the paper is required and some more realistic benchmarks would be nice.
* Maybe I have misunderstood the variational warm-up procedure. But using the MLE estimates to initialize the BNN mean parameters was done in VCL https://arxiv.org/abs/1710.10628 . So this is not a novel idea. Furthermore, there are better ways to initialize a BNN such as using Bayesian linear regression: https://proceedings.mlr.press/v97/rossi19a/rossi19a.pdf.

Clarity
* Why does the teacher model use EMA updates instead of using the inference variational posterior?
* Why is data augmentation an important component in VCoTTA (Sec 4.2)? This is suddenly presented in the paper without justification.

Notation:
* In Section 4.2, the title is a “Mixture-of-Gaussian prior”, but Eq 11 is an addition of two priors which are Gaussians by design, so this is a “scale-mixture prior” https://arxiv.org/pdf/1505.05424, not a mixture of Gaussians (https://www.inf.ed.ac.uk/teaching/courses/mlpr/2016/notes/w9b_mixture_models.pdf)?
* Confusing notation of the source prior: it is denoted as $p_0$ (Fig 2) and $p_1$ (Eq 11), this needs to be consistent.


Empirical weaknesses:
* No standard errors in the experimental results. So difficult to see which method outperforms another.
* Uncertainty estimation is not performed with standard methods like ECE or OOD detection like https://arxiv.org/abs/2102.06571. It is unclear to me whether the Brier Score estimates uncertainties.

**Questions:**

Please see the weaknesses above. Additionally, there are some other minor questions mainly regarding the writing and the questions around implementation:
* Line 169: what is a “serious data augmentation”?
* Line 178: what is a confidence function?
* Eq 13: what is $x’$?
* Alg 1, line 2: how do you initialize the prior distribution $p(\theta)$ with the MLE weights from a pre-trained CNN? Do you set the mean variational means to the MLE values and initialize the variances to be some small initial value such as $10^-3$?
* On the first domain adaptation update of VCOTTA, how is $\bar{p}_1$ defined since we do not have an initial teacher model yet (Eq 11)?

**Limitations:**

There is a good discussion on the limitations of VCoTTA.

One limitation that is not discussed is in the effectiveness of (variational) Bayesian sequential inference methods. Weight space variational inference has been shown to be very difficult to do in practice, https://arxiv.org/abs/2301.01828. So weight space variational inference (without tricks like multi-head networks and coresets) might not be the best choice when wanting to remember a source distribution.

---

> ### Author Rebuttal · Authors · 2024-08-06
>
> ### Weakness of Noverty
>
> **1. CIFAR10 dataset is not enough**
>
> **Response**: Many CTTA methods focus on classification problems because validating classification on foundational and well-recognized datasets can reduce potential confounding factors and more accurately assess the effectiveness of the proposed methods. Many applications can be viewed as classification problems, such as semantic segmentation. We did not only conduct experiments on CIFAR10. In Tab. 2&3, we validated on CIFAR100 and ImageNet, which are generally considered more complex than CIFAR10. Many existing CTTA works have been conducted on basic classification tasks such as PETAL and RoTTA, which demonstrates that our experiments are sufficient. Given the very short rebuttal period, it is challenging to immediately validate the method's effectiveness in specific real-world scenarios, but we find your suggestion valuable and will incorporate it in the future.
>
> **2. BNN initialization**
>
> **Response**:
>
> (1) The variational warm-up (VWU) strategy aims to obtain a suitable variational distribution on the source data, and is based on a pre-trained CNN. Of course we can train a BNN from scratch, but the training is difficult. VWU strategy is more like a pre-trained trick instead of BNN initialization.
>
> (2) The initialization used by VCL is *"the prior $N (0, I)$ and ... a very small initial variance (10^−6)"*, which is a direct initialization. However, VWU is different from the initialization method. The significant difference is that we require a complete pre-trained CNN, and the advantage of VWU can be seen in Appendix F. In Tab. 10, we have compared with common initialization and find that VWU is more convenient。
>
> (3) [1] also aims to convert a pre-trained model into a BNN but depends to complicated model design. Instead, we prefer a simple reparameterization trick to solve this problem, which is to quickly offer a pretrained source BNN model for the following TTA algorithm.
>
> (4) We have compared VWU with random initialization in Tab. 10. We further compare VWU with VCL initialization in **Table 8 in the attached PDF** and also find similar performance.
>
> 	[1] Make Me a BNN: A Simple Strategy for Estimating Bayesian Uncertainty from Pre-trained Models. CVPR. 2024.
>
>
> ### Weakness of Clarity
>
> **1. Why does the teacher use EMA?**
>
> **Response**: In the mean-teacher (MT) structure, a student model is trained on unlabeled data and receives guidance from a teacher model. The teacher model is an exponential moving average (EMA) of the student model, provides stable pseudo-labels for the unlabeled data. EMA smooths the updates to the teacher model, reducing fluctuations. This stability helps prevent the student model from being misled by error accumulation, promoting more robust learning.
>
> **2. Why is data augmentation an important component?**
>
> **Response**: A large pool of augmentations and averaging them would give more robust evaluation. In Eq.10, we use the mean entropy derived from data augmentations to represent the confidence of the two prior models, and mix up the two priors with a modulating factor. Each addition item is a softmax with one type of augmentation. The softmax is to given the confidence of the source and teacher models. Eq.13 is an improved version for teacher log-likelihood in Eq.9, which picks up the augmented logits with a larger confidence than the raw data with $\epsilon$ margin. We have conducted the ablation study on augmentation in Appendix E.2, Tab.9. Tab.9 shows that increasing the number of augmentations can enhance effectiveness.
>
> ### Weakness of Notation
>
> **1. "scale-mixture prior"?**
>
> **Response**: The title "Mixture-of-Gaussian prior" is a simplification of "a scale mixture of two Gaussian densities as the prior" as description in the mentioned paper. Thanks to the suggestion, we will add a description and citation in the section to distinguish our method from the Gaussian Mixture Model method to avoid confusion.
>
> **2. p0 (Fig 2) and p1 (Eq 11)**
>
> **Response**: This is a typo error, we will revise this.
>
> ### Weakness of Experiment
>
> **1. No standard errors**
>
> **Response**: We have provide the standard error comparison in Appendix H, Table 11.
> In Table 11, we conduct 10 different orders, and show the average performance of all compared methods. The proposed VCoTTA outperforms other methods on the standard error of three dataset, which shows the effectiveness of the prior calibration in CTTA.
>
> **2. Why use BS to evaluate uncertainty**
>
> **Response**: BS is a well-recognized uncertainty estimation method, quantifying the MSE between predicted probabilities and actual outputs. [1] states that "one of the first metrics ... widely used ... is Brier score (BS)". BS has been widely used in many uncertainty estimation such as [1-4]. We also evaluate uncertainty using ECE metric, and the results can be seen in **Table 9 in the attached PDF**, and we can find similar results as BS.
>
> 	[1] Better Uncertainty Calibration via Proper Scores for Classification and Beyond. NeurIPS 2022.
> 	[2] A probabilistic framework for lifelong test-time adaptation. CVPR 2023.
> 	[3] Calibration of neural networks using splines. ICLR 2020.
> 	[4] Intra order-preserving functions for calibration of multi-class neural networks. NeurIPS 2020.
>
>
> ### Question 1: Line 169: what is a “serious data augmentation”?
>
> **Response**: This should be "a series of data augmentation". We will revise this.
>
> ### Question 2: Line 178: what is a confidence function?
>
> **Response**: Max value of softmax output.
>
> ### Question 3: Eq 13: what is x′?
>
> **Response**: x is the raw test data while x' is its augmented version.
>
> ### Question 4: Alg 1, line 2, initialize the variances to be some small?
>
> **Response**: Yes, in VWU, we initialize the varizances with log(1+e^-4)
>
> ### Question 5: how is $\bar{p}_1$ defined?
>
> **Response**: The teacher model is initialized to be the same as the student model and will be different from the student after updating.

---

> > ### Comment · Reviewer_yBx1 · 2024-08-12
> > **Response to rebuttal**
> >
> > Thanks for taking the time to respond to my queries.
> >
> > ## BNN initialization
> >
> > > The initialization used by VCL is "the prior $N(0, 1)$ and ... a very small initial variance (10^−6)", which is a direct initialization. However, VWU is different from the initialization method. The significant difference is that we require a complete pre-trained CNN, and the advantage of VWU can be seen in Appendix F. In Tab. 10, we have compared with common initialization and find that VWU is more convenient.
> >
> > Yes, the prior is $N(0, 1)$, but the variational mean parameters are initialized by using a network with the same architecture pre-trained using MLE. This might not be stated explicitly, but is a feature of the implementation and a common feature of variational BNNs.
> >
> > ## Overall
> >
> > I will raise my score, but still do not think this is ready for publication. I would like to see a more real-world scenario beyond classification problems to really show case TTA. I would also like to see standard errors by default on most if not all experiments.

---

### Official Review · Reviewer_9v84 · 2024-07-04

**Soundness:** 3
**Presentation:** 2
**Contribution:** 3
**Rating:** 6
**Confidence:** 2

**Summary:**

This paper introduces VCoTTA, a novel variational Bayesian approach to address the Continual Test-Time Adaptation (CTTA) task, which focuses on effective domain adaptation during continuous domain shifts at test time. The authors' main contributions include a method to measure uncertainties in CTTA, addressing the issue of error accumulation due to the use of unlabeled samples. They propose transforming a pretrained deterministic model into a Bayesian Neural Network (BNN) using variational warm-up at the source stage, and employ a mean-teacher update strategy during test time. The approach updates the student model by combining priors from both source and teacher models, with the evidence lower bound formulated as the cross-entropy between student and teacher models, along with the Kullback-Leibler (KL) divergence of the prior mixture. Experimental results on three datasets demonstrate the method's effectiveness in mitigating error accumulation within the CTTA framework.

**Strengths:**

The paper demonstrates originality through the novel Variational Continual Test-Time Adaptation (VCoTTA) approach, which creatively utilizes Bayesian Inference for Continual Test-Time Adaptation, and employs strategies like the variational warm-up and prior mixture techniques. The quality of the work is evident in its solid theoretical foundation, comprehensive methodology, and empirical validation on multiple datasets.

The paper's clarity is apparent in its well-structured presentation, use of visual aids, and explicit statement of contributions. The significance of the research is underscored by its practical relevance to risk-sensitive applications, potential for broad applicability, and the reported improvements in predictive accuracy and uncertainty estimation under distribution shifts. By addressing critical challenges in CTTA, such as error accumulation and uncertainty estimation, and bridging Bayesian methods with test-time adaptation, the paper not only advances the current state of the art but also opens up promising avenues for future research. Overall, this work represents a valuable contribution to the field, offering both theoretical insights and practical advancements in continual learning and test-time adaptation.

**Weaknesses:**

Regarding the computational overhead discussed in the paper, while it is noted that online Variational Inference is employed to make the approach computationally feasible, a detailed analysis of the computational costs associated with VCoTTA is absent. Table 13 presents a comparison of time and memory costs, but the source of these values is unclear. Could you specify which dataset was used for these measurements? Also, is it possible to clarify whether the time and memory comparisons pertain to training or testing phases?

The manuscript contains several typographical and grammatical errors that need to be addressed. Specifically, brackets are missing in Equation (5) and in the sentence following Equation (13). Could these omissions be corrected to prevent misinterpretation of the mathematical expressions and enhance the clarity of the paper?

There are multiple grammatical issues that require rectification. The sentence "MT is initially proposed in semi-supervised and unsupervised learning" is somewhat unclear. Could this be rephrased for better coherence? Additionally, the sentence "We use the mean entropy derived from a given *serious* data augmentation to represent the confidence of the two prior models, and mix up the two priors with a modulating factor" appears to contain a typographical error and could be better structured. Could these issues be addressed to improve the readability and accuracy of the text?

The heading for Section 5.7 seems to not accurately reflect the content discussed within. Could this heading be revised to more accurately convey the main topics or findings of the section, thereby ensuring clarity and relevance for the reader?

The explanation of how MT operates in semi-supervised and unsupervised learning settings appears incomplete and potentially misleading. The current statement, "where the teacher model guides the unlabeled data, helping the model generalize and improve performance with the utilization of large-scale unlabeled data," lacks specificity. Could you specify which model (teacher or student) benefits from this guidance and in what manner? Additionally, the phrase "where the teacher model guides the unlabeled data" seems incorrect. Could this be clarified or corrected to accurately reflect the operational dynamics of the MT framework?

**Questions:**

Please provide answers to the questions/suggestions provided in the Weaknesses section.

**Limitations:**

The authors have adequately addressed the limitations in the paper.

---

> ### Author Rebuttal · Authors · 2024-08-06
>
> ### Weakness 1: Further analysis on efficiency
>
> **Response**:
> Our approach incorporates a Variational Warm-Up (VWU) strategy during pretraining and utilizes VCoTTA for test-time adaptation. We conduct additional cost analyses under various settings, including different batch sizes and model sizes. The detailed cost results are provided in **Tables 4 and 5 of the attached PDF** for both the warm-up and test phases. The results indicate that while the VWU strategy becomes more efficient with an increase in batch size, this also leads to higher memory consumption. Similarly, during the test phase, batch size has a similar effect, and the efficiency is significantly influenced by the model size too. It is important to note that CTTA comprises two primary components: testing and adaptation. The use of Bayesian methods does not impact testing efficiency, as variance is not actively engaged during inference.
>
> ### Weakness 2: Typographical and grammatical errors
>
> **Response**: Thank you for your valuable suggestion. We will thoroughly review the paper for any typographical and grammatical errors and make revisions to improve clarity and readability. Additionally, we will include more references to substantiate the accuracy of our claims.
>
> ### Weakness 3: Revision suggestion on the heading of Sec. 5.7.
>
> **Response**: Thank you for your suggestion. Since Section 5.7 presents experiments on continual domains with gradually changing shifts, we will update the heading to "Comparisons under Gradually Changing Domain Shifts".
>
> ### Weakness 4: Mean-teacher statement clarification
>
> **Resonse**:
>
>  (1) *"where the teacher model...." lacks specificity*: The mean-teacher (MT) structure is a method in semi-supervised and unsupervised learning where a student model, trained on both labeled (if semi-supervised) or unlabeled data, receives guidance from a teacher model. The teacher model, which is an exponential moving average (EMA) of the student model’s weights, provides stable pseudo-labels for the unlabeled data. This setup encourages the student model to produce consistent predictions and effectively leverage large-scale unlabeled data, improving generalization and overall performance by stabilizing training and enhancing learning from broader data distributions.
>
> (2) *Which model (teacher or student) benefits from this guidance and in what manner*: In the mean-teacher structure, the **student model** benefits from the guidance provided by the **teacher model**. The teacher model is an EMA of the student model's weights. These pseudo-labels guide the student model during training by providing consistent targets for the unlabeled data, which helps the student model to generalize better and improve its performance. This process ensures that the student model learns more robust features from the unlabeled data, enhancing its ability to make accurate predictions on new, unseen data.
>
> (3) *"where the teacher model guides the unlabeled data" seems incorrect*: We will revise the explanation of the mean-teacher structure in our paper and provide more reference to support our statement. Thank you for your suggestion!

---

### Official Review · Reviewer_p6Bz · 2024-07-09

**Soundness:** 2
**Presentation:** 2
**Contribution:** 2
**Rating:** 5
**Confidence:** 3

**Summary:**

The paper proposes a method to continually adapt a pre-trained classifier to an unlabeled stream of test data. They address the problem of continual test-time adaptation through the lens of Bayesian deep learning. Their method consists of three main components: (1) a variational warm-up strategy to turn any source model into a Bayesian Neural Network, (2) a mixing strategy between the source model and the last posterior to leverage the trade-off between adaptation and forgetting, and (3) a modified entropy term that is symmetric and incorporates data augmentations. The authors compare their method on standard CIFAR-C and Imagenet-C datasets to a set of TTA baselines. The paper also includes ablation studies on the components and an evaluation of uncertainty estimates.

**Strengths:**

- The paper tackles one of the most relevant problems in continual domain adaptation, namely the trade-off between agile adaptation while preventing the forgetting of the source model. It does so by constituting a mixture of the source and last adapted model in a VI framework, which is novel to my knowledge. (originality)
- Further, in a setting where robustness is crucial and therefore uncertainty quantification can be helpful, the combination of Bayesian deep learning and continual test-time adaptation is interesting and insightful. (originality)
- The methodological backbone is accompanied by insightful ablation studies that highlight the significance of the different parts of the paper’s contribution. (quality)
- The paper is, in most parts, pleasant to read. The notation is clear and consistent, and the reader is well guided through the different sections. (clarity)
- The paper presents clear experimental evidence in support of the method. The experiments show an improvement in adaptation accuracy on already quite saturated datasets (up to 1.8% points on CIFAR-10-C). VCoTTA also seems to be advantageous on most corruption types. (significance)

**Weaknesses:**

The paper presents strong evidence in support of the proposed method. However, it is left unclear to me why the method performs so much better than previous approaches.

- The method consists of a range of specific components. However, in some cases, the specific design of the components is not clearly motivated. In particular, equations 10 and 13 lack supporting citations or explanations. Why have exactly these formulations been chosen?
- I’d like to get more clarity on the difference between this paper and the original CoTTA work, as it seems to me there are certain components in common (e.g., student-teacher approach, EMA). More precisely, could you please highlight the difference in the update equations between the two papers? My understanding is that adding the VI framework notably changes (i) the optimization objective by adding the KL term (instead of solely minimizing entropy) and (ii) the predictions by marginalizing out the model parameters. Where else does the VI framework contribute to differences?

**Questions:**

- Recent work has explored the degradation of CTTA methods after a long adaptation period [1]. Given that the paper argues that the mixed prior between the adapted and source model prevents forgetting, it would be interesting to see for how long the adaptation can be successful. Is there any experimental evidence on the robustness over time of the adaptation method? See [1] for an example of a suitable benchmark dataset.
- What is the motivation behind Equation 13? In other words, why would we only want to have log-likelihood terms of “confident augmentations”? I would have assumed that having a large pool of augmentations and averaging them all would give more robust predictions. An ablation study on the design of this term would also be insightful.
- Could the method suffer from mode averaging in Equation 11?
- Am I assuming correctly that only one gradient step is performed per test batch?

[1] Press, Ori, et al. "Rdumb: A simple approach that questions our progress in continual test-time adaptation." Advances in Neural Information Processing Systems 36 (2024).

**Limitations:**

The authors have listed limitations including computational efficiency and the need for access to the source data at adaptation time

---

> ### Author Rebuttal · Authors · 2024-08-06
>
> ### Weakness 1: Why superior to SOTA
>
> **Response**: Our method outperforms the SOTA approaches because it leverages the BNN ability to estimate model uncertainty, which reduces error accumulation from continual unknown domains during the testing phase. We find that the unreliable priors may affect the performance of BNNs in the CTTA task. To address this, we utilize variational inference to compute the Evidence Lower Bound (ELBO), and propose to improve the calculation of the Entropy and KL terms. For the Entropy term, we propose using a Mean-Teacher (MT) structure to transform the original conditional entropy into cross-entropy, taking advantage of MT's delayed update characteristic. For the KL term, we introduce a Gaussian mixture prior enhancement method that directly reduces the impact of unreliable priors. Additionally, the variational weight uncertainty strategy enables the model to have some before-test uncertainty estimation capability. These modules allow the proposed Bayesian method to mitigate the influence of unreliable priors in CTTA tasks, leading to better performance. All of these modules are explained in detail in the paper, and ablation experiments are provided. If there are any unclear aspects, please feel free to ask further questions.
>
> ### Weakness 2: Motivations of our design
>
> **Response**: The key components and workflow as been shown in the above response. We further explain Eq. 10 and Eq. 13 as follows. In Eq 10, we use the mean entropy derived from a given series of data augmentation to represent the confidence of the two prior models, and mix up the two priors with a modulating factor. Each addition item is a simple softmax with one type of augmentation, and $\mathcal{I}$ denotes the augmentation types. The softmax is to given the confidence of the source model and $1-\alpha$ means the confidence of the teacher model. Eq. 13 is an improved version for teacher log-likelihood in Eq. 9. Eq. 13 picks up the augmented logits with a larger confidence than the raw data with $\epsilon$ margin.  Eq. 13 can be regarded as a filter, meaning that for each sample, the reliable teacher is represented by the average of its augmentations with $\epsilon$ more confidence than the raw data's.
>
> ### Weakness 3: Comparison with CoTTA
>
> **Response**:
>
> *(1) BNN for CTTA task*: First, BNNs offer several advantages over traditional CNNs, especially in scenarios where uncertainty estimation, robustness to overfitting, and the ability to incorporate prior knowledge are important. BNNs provide several features that are beneficial for test-time adaptation, including uncertainty estimation, robustness to overfitting, the incorporation of prior knowledge, adaptive complexity, and enhanced interpretability. These advantages enable BNNs to adapt more effectively when encountering new, unseen, or uncertain data during test-time, making them well-suited for dynamic and evolving environments. **However**, directly using existing BNN for CTTA is ineffective because of the unreliable prior, and our goal is to reduce the influence of unreliable prior. See more detail in the response of Weakness 1. Some works using BNN for traditional TTA task can be found in [1-3].
>
> *(2) CoTTA vs. Ours*: In comparison with CoTTA, we have the following difference：
>
> 1. CoTTA is based on CNN while our method is based on BNN, and the advantage of BNN for CTTA task can be seen above.
> 2. CoTTA focuses on the error accumulation and catastrophic forgetting when using CNN. Our method solves the unreliable prior issue under BNN structure, which may lead to error accumulation and catastrophic forgetting.
>
> *(3) About the reviewer's understanding*: The objective the reviewer mentioned is the ELBO (Eq. 6) of Variational Inference (VI) in CTTA, which is derived from the VI assumption. VI assumes that there exists a variational distribution $q(\theta)$ that approximates the true posterior $p(\theta|\mathcal{U})$ . The approximation process can be represented by a KL divergence optimization (Eq. 5). However, it is difficult to directly optimize the KL divergence, the ELBO is an alternative for optimization. For prediction, the BNN can be reduced to a CNN because the variance is not used
>
> 	[1] Extrapolative continuous-time bayesian neural network for fast training-free test-time adaptation. NeurIPS, 2022.
> 	[2] Bayesian adaptation for covariate shift. NeurIPS, 2021.
> 	[3] Task-agnostic continual learning using online variational bayes with fixed-point updates. Neural Computation, 2021.
>
> ### Question 1: Long adaptation period
>
> **Response**: We have evaluated on long-adaptation period in our Appendix I, Fig. 6. In Fig. 6, we evaluate on 10 loops of a same corruption order, which means we have 15 domain shift repeat 10 times, yielding 150 domain shifts, we believe that 150 domain shifts is a long adaptation period. The results show that most compared methods obtain performance improvement in the first several loops, but suffer from performance drop in the following loops. This means that the model drift can be even useful in early loops, but the drift becomes hard because of the unreliable prior. The results also indicate that our method outperforms others in this long-term adaptation situation and has only small performance drops.
>
> ### Question 2: Motivation and ablation study on Eq. 13
>
> **Response**: The motivation of Eq.13 can be seen in the response of Weakness 2. We have conducted the ablation study in Appendix E.2, Table 9. Table 9 shows that increasing the number of augmentations can enhance effectiveness, but this hyperparameter ceases to have a significant impact after reaching 32.
>
> ### Question 3: Mode averaging
>
> **Response**: We guess the reviewer means "model averaging", i.e., set $\alpha=0.5$ for Eq. 11. Yes, this has been verified in Table 5 in the submitted manuscript. We may misunderstand the question, and the reviewer can give further description.
>
> ### Question 4: One gradient step
>
> **Response**:  Yes, your are right.

---

> > ### Comment · Reviewer_p6Bz · 2024-08-12
> >
> > >**Weakness 2: Motivations of our design**
> > >
> >
> > I appreciate the additional comments on Equations 10 and 13. I would like to see them included in the revised version. I believe that providing a more detailed explanation of how these design choices are motivated would strengthen the paper.
> >
> > >**Weakness 3: Comparison with CoTTA**
> > >
> >
> > Thank you for the response. I believe I understand the high-level differences between the CoTTA and VCoTTA papers as provided in your response. However, my question was aimed at the lower-level distinctions between the two approaches. Despite the high-level difference between the two (e.g., CNN vs. BNN perspective), I do believe there are many lower-level similarities. In my question, I’d like to get clarity about the low-level differences (see the example of the student loss below). I feel this has not yet been addressed.
> >
> > > *(3) About the reviewer's understanding*: The objective the reviewer mentioned is the ELBO (Equation 6) of Variational Inference (VI) in CTTA, which is derived from the VI assumption. VI assumes that there exists a variational distribution q(θ) that approximates the true posterior p(θ|U). The approximation process can be represented by a KL divergence optimization (Equation 5). However, it is difficult to directly optimize the KL divergence, so the ELBO is an alternative for optimization. For prediction, the BNN can be reduced to a CNN because the variance is not used.
> > >
> >
> > My description might have been misleading. I am referring to the student loss of VCoTTA (Equation 15) and comparing it to the student loss of CoTTA (Equation 1 in [1]). I think one can see an example of a lower-level difference there (i.e., the additional term in VCoTTA, $\alpha \times KL (q_t||q_0) + (1 − \alpha) \times KL (q_t|| \bar{q}_t)$).
> >
> > I’d like to get a better overview of these kinds of differences (i.e., differences in the *equations*). I think this could provide the reader with a better understanding of why the method performs so well (weakness 1).
> >
> > [1] Continual Test-Time Domain Adaptation, CVPR, 2022.
> >
> > >**Question 1: Long adaptation period**
> > >
> > Thank you for pointing to the additional experiment. This answers my question.
> >
> > >**Question 2: Motivation and ablation study on Equation 13**
> > >
> > Thank you for the clarifications. My question has been answered.

---

> ### Author Response · Authors · 2024-08-12
>
> Thank you for your comments, we response to the difference of low-level distinctions between CoTTA and VCoTTA as follows.
>
> 1. **CoTTA Contributions and Equations**: First, let's review the three main *contributions of CoTTA* and then explain the *equation differences* between our approach and CoTTA in regard to the 3 contributions and 8 equations. Note that Eq. 1, 2 and 5 are the updates for student and teacher models.
>
> 	- *Weight-Averaged Pseudo-Labels*: This contribution is in fact the Mean-Teacher (MT) structure use in CTTA. We also adopt the MT structure like CoTTA’s design. The equation differences are:
> 		- CoTTA Eq. 1&5: Student update. The reviewer has mentioned this, and our further explanation is provided below (Response 2).
> 		- CoTTA Eq. 2: Teacher update. Both CoTTA and VCoTTA use EMA for updating, but VCoTTA is based on BNNs (Our Eq. 16), where the BNN is updated in Gaussian distribution.
> 	- *Augmentation-Averaged Pseudo-Labels* (CoTTA Eq. 3&4): CoTTA chooses to use either augmented or non-augmented teacher pseudo-labels based on whether the source model's confidence in the sample exceeds a certain threshold. In our method, we do not make such a judgement. For the augmentation, we first leverage data augmentation to calculate the mixing coefficient between the teacher prior and the source prior (Our Eq. 10), then enhance the teacher prior (Our Eq. 13) as we mentioned in the rebuttal.
> 	- *Stochastic Restoration* (CoTTA Eq. 6&7&8): CoTTA randomly restores part of student model parameters to the corresponding parameters in the source model with a certain probability to preserve the knowledge from the source. Our method does not employ this stochastic restoration strategy. Our mixture of priors (Our Eq. 11,12,15) may have similar form as CoTTA Eq. 8, but we do not restore student model using source model randomly but mix the teacher prior and source prior with an adaptive factor $\alpha$.
>
> 2. **Difference in Student loss**: The reviewer has pointed out the student loss difference in low-level. Compared to CoTTA, the main difference in student loss is the KL term at the end of our Eq.15. The KL term can be considered a regularization constraint on the model parameters. $q_t$ , $q_0$ and $\bar{q}_t$ represent the parameter distributions of the student model (to be optimized), source model, and augmented teacher model, respectively. In our method, all three distributions are Gaussian. The calculation of the KL term for Gaussian distributions can be referenced in Eq. 25 of our appendix, which can be computed directly in closed form. The meaning of the KL term is to provide constraints on the student model update when faced with unreliable priors, using both the source model and teacher model. These two constraints are controlled by the weights calculated in our Eq. 10.
>
> Thank you to the reviewer for pointing out that we need to improve our method presentation. We will enhance the clarity of our descriptions in the paper to make the meaning of the formulas easier for readers to understand.

---

> > ### Comment · Reviewer_p6Bz · 2024-08-14
> >
> > Thank you for very much for providing this overview. It addressed the concern I raised initially. I raised my score to 5.

---

### Official Review · Reviewer_st3m · 2024-07-12

**Soundness:** 3
**Presentation:** 3
**Contribution:** 3
**Rating:** 6
**Confidence:** 3

**Summary:**

The paper presents a variational Bayesian approach to handle uncertainties in continual test-time adaptation (CTTA). The source pretrained model is made Bayesian by variational warm and a mean-teacher update strategy is used at test time. To avoid drift due to uncertainty of priors using only unlabeled data at test time, the paper proposed to update the student model by combining priors from both the source and teacher models. The evidence lower bound is formulated as the cross-entropy between the student and teacher models, along with the Kullback-Leibler (KL) divergence of the prior mixture. Experimental results on three datasets demonstrate the method’s effectiveness in mitigating error accumulation within the CTTA framework.

**Strengths:**

- Novelty: Bayesian approach in Continual Learning is a principled and elegant approach to the problem which this paper is relying on. In CTTA, there are additional issues due to the uncertainty of the prior distributions using only the unlabeled data from unknown domains. This paper presents a novel solution by using adaptive mixture prior models and student-teacher update on top of an existing framework.

- Relevance: CTTA is a topic that can interest a general audience, and the  Bayesian and variational framekwork can also be of interest to many.

**Weaknesses:**

- While Bayesian approach is nice in principle, it can be computationally demanding and offer little benefit in practice. Most of the existing CTTA methods are computationally and memory efficient, whereas this method present an opposite end of the spectrum. While the reported results are impressive, it is unclear to me why the proposed method is superior to other SOTA methods.
- The hyperparameter selection process is not addressed in the paper, which is critical in TTA where all hyperparameters should be predetermined before data access. How are they chosen?

**Questions:**

- What is the rough analysis of the space and time complexity of the proposed approach, separate for off-line and online phases? There is only one incomplete empirical comparison in the appendix, which leave me curious how the space and time cost change in different settings (e.g., batch size, training set size, model size, etc).

- In the experiments, can resetting to source simply solve the drift issue and achieve much better results than the naive results in the tables?

- For the mixture rate alpha, what is the best result with a non-adaptive alpha?

**Limitations:**

Limitations are mentioned in the paper, albeit very brief. Overall, the proposed method is more complex and demanding (such as requiring a pretrained probabilistic model or source data) for TTA applications. Perhaps the proposed approach may work even better with UDA or other CL scenarios with (partially available) target labels?

---

> ### Author Rebuttal · Authors · 2024-08-06
>
> ### Weakness 1: Bayesian approach for CTTA and why superior to SOTA
>
> **Response**:
>
> (1) *Bayesian approach for  CTTA*
>
>  Bayesian networks have already been applied in the field of TTA task. For example, [1] develops a continuous-time Bayesian neural networks to process non-stationary streaming data in real-time. [2] provides for a well-defined relationship between unlabeled inputs under distributional shift and model parameters based on Bayesian method at test-time. BayTTA [3] designs a Bayesian model for medicine TTA task. Actually, Bayesian networks can also be applied to online learning. [4] proposes a Bayesian-inference based recommendation system for online social networks. [5] is based on Bayesian networks, and is assumed to transition smoothly in the joint space of numerical parameters and graphical topology, allowing for robust online network learning.
>
> In CTTA task, in addition to efficiency, stable testing and adaptation are also important. A CTTA model may suffer from error accumulation due to uncertainties arising from multiple domain shifts over extended periods. Bayesian methods can mitigate this issue by estimating uncertainty. CTTA involves two main parts: testing and adaptation. The use of Bayesian methods does not impact testing efficiency, as variance is not actively engaged during inference.
>
> 	[1] Huang H, et al. Extrapolative continuous-time bayesian neural network for fast training-free test-time adaptation. NeurIPS, 2022.
> 	[2] Zhou A, et al. Bayesian adaptation for covariate shift. NeurIPS, 2021.
> 	[3] Sherkatghanad Z, et al. BayTTA: Uncertainty-aware medical image classification with optimized test-time augmentation using Bayesian model averaging[J]. arXiv preprint arXiv:2406.17640, 2024.
> 	[4] Yang X,  et al. Bayesian-inference-based recommendation in online social networks. IEEE TPDS, 2012.
> 	[5] Wang Z, et al. Time varying dynamic Bayesian network for nonstationary events modeling and online inference. IEEE TSP, 2010.
>
> (2) *Why superior to SOTA*
>
> Our method outperforms the SOTA approaches because it leverages the BNN ability to estimate model uncertainty, which reduces error accumulation from continual unknown domains during the testing phase. We find that the unreliable priors may affect the performance of BNNs in the CTTA task. To address this problem, we utilize variational inference to compute the Evidence Lower Bound (ELBO), and propose to improve the calculation of the Entropy and KL terms.
> For the Entropy term, we propose using a Mean-Teacher (MT) structure to transform the original conditional entropy into cross-entropy, taking advantage of MT's delayed update characteristic. For the KL term, we introduce a Gaussian mixture prior enhancement method that directly reduces the impact of unreliable priors. Additionally, the variational weight uncertainty strategy enables the model to have some before-test uncertainty estimation capability.
> These modules allow the proposed Bayesian method to mitigate the influence of unreliable priors in CTTA tasks, leading to better performance. All of these modules are explained in detail in the paper, and ablation experiments are provided. If there are any unclear aspects, please feel free to ask further questions.
>
>
> ### Weakness 2: Bayesian approach for CTTA and why superior to SOTA
>
> **Response**:
>
> In our paper, we have evaluate several highly related hyperparameters including confidence margin in Table 8, augmentation number in Table 9, warm-up epochs in Fig. 4, warm-up data scale in Fig. 5. We further provide some other hyperparameters in the attached tables including learning rate, batch size and the softmax temperature.
>
> (1) Learning rate: See **Table 1 in the attached PDF**.
> (2) Batch size: See **Table 2 in the attached PDF**.
> (3) Softmax temperature: See **Table 3 in the attached PDF**.
>
> ### Question 1: Space and time cost of offline and online phase
>
> **Response**: We provide more cost experiments under different setting including different batch size of different model size. The detail cost values can be seen in **Table 4 and 5 in the attached PDF**.
>
> ### Question 2: Reseting to source model
>
> **Response: In CTTA task , model does not know when the domain shift happen. Thus, resetting model to source model is not feasible in CTTA task. We conduct this experiment on CoTTA and our method, the results can be seen in **Table 7 in the attached PDF**.
> We find that the resetting performance is less effective than the continual setting on both CoTTA and our method. The results show that the existing datasets may have shared knowledge across domains.
>
> ### Question 3: Best non-adaptive $\alpha$
>
> **Response**: We further evaluate on more $\alpha$, and the results can be seen in **Table 6 in the attached PDF**. The best non-adaptive $\alpha$ is 0.7. The results show that setting a fixed $\alpha$ is not effective enough for CTTA task. This underscores the significance of striking an adaptive balance between the two prior models in an unsupervised environment. The trade-off implies the need to discern when the source model’s knowledge is more applicable and when the teacher model’s shifting knowledge takes precedence.

---

> > ### Comment · Reviewer_st3m · 2024-08-08
> >
> > Thank you for a detailed response to my questions.
> > - The additional results are helpful to understand the complexity of the approach and the model selection better.
> > - The answer to why it's superior to SOTA is still circumstantial but I understand that it's not an easy question.
> > - Speaking of SOTA, it came to my attention that the experiments are missing comparisons with ViDA [1] and EcoTTA [2] which are frequently cited in other papers. Any reason why?
> >
> >
> > [1] Liu, Jiaming, et al. "Vida: Homeostatic visual domain adapter for continual test time adaptation." arXiv preprint arXiv:2306.04344 (2023).
> > [2] Song, Junha, et al. "Ecotta: Memory-efficient continual test-time adaptation via self-distilled regularization." Proceedings of the IEEE/CVF Conference on Computer Vision and Pattern Recognition. 2023.

---

> > > ### Author Response · Authors · 2024-08-12
> > >
> > > Thank you for your further comments, and our responses are as follows.
> > >
> > > **A brief description of the effective reason**: To illustrate why it is effective, we simplify the description of our key idea: The characteristics of BNN make it more effective in uncertain testing scenarios, but directly using BNNs in CTTA may lead to error accumulation due to unreliable priors. To address these unreliable priors, we propose the VCoTTA method, which enhances the performance of VI in CTTA tasks by enhancing the priors in a Mean-Teacher structure.
> > >
> > > **Two other SOTA methods to compare**: Thank you for your kind suggestion. Due to space constraints in the manuscript, we choose some other SOTA methods for comparison such as PETAL (CVPR24), which is more related to our motivation. We show the comparison results on CIFAR10C with the two methods in the table below, and we will to include these comparisons in the revised manuscript.
> > >   | Method | Avg. err |
> > >   |---|---|
> > >   | ECoTTA  | 16.8 |
> > >   | Vida | 15.8 |
> > >   | Ours  | 13.1 |

---

> ### Comment · Reviewer_st3m · 2024-08-13
>
> Thank you for the additional results. These resolved some additional concerns I had about the paper. I am raising my score from 5 to 6.

---

### Author Rebuttal · Authors · 2024-08-06

Dear Reviewers:

We thank the reviewers for their careful examination of our paper and for providing a wealth of valuable suggestions. We also appreciate the reviewers' recognition of our work in terms of originality, relevance, quality, clarity, and significance. **The attached PDF contains the mentioned experimental results.**
We focus our response on several key concerns:

**Why use BNN for CTTA task?**

**Response from the authors**: Reviewers st3m, p6Bz and 9v84 all appreciated the innovation of using a variational inference framework in CTTA and agree the originality. However, as we list in the limitation, the BNN may have less efficiency than CNN. In our method, we focus on stable testing and adaptation, which are also important. A CTTA model may suffer from error accumulation due to uncertainties arising from multiple domain shifts over extended periods. Bayesian methods can mitigate this issue by estimating uncertainty. CTTA involves two main parts: testing and adaptation. The use of Bayesian methods does not impact testing efficiency, as variance is not actively engaged during inference. Moreover, Bayesian networks have already been applied in the field of TTA task as we mentioned in the response to Reviewer st3m.

**Why is it better than SOTA?**

**Response from the authors**: Reviewers p6Bz and 9v84 found that the proposed method outperforms the SOTA methods. Our method outperforms the SOTA approaches because it leverages the BNN ability to estimate model uncertainty, which reduces error accumulation from continual unknown domains during the testing phase. We find that the unreliable priors may affect the performance of BNNs in the CTTA task. To address this problem, we utilize variational inference to compute the Evidence Lower Bound (ELBO), and propose to improve the calculation of the Entropy and KL terms. For the Entropy term, we propose using a Mean-Teacher (MT) structure to transform the original conditional entropy into cross-entropy, taking advantage of MT's delayed update characteristic. For the KL term, we introduce a Gaussian mixture prior enhancement method that directly reduces the impact of unreliable priors. Additionally, the variational weight uncertainty strategy enables the model to have some before-test uncertainty estimation capability. These modules allow the proposed Bayesian method to mitigate the influence of unreliable priors in CTTA tasks, leading to better performance. All of these modules are explained in detail in the paper, and ablation experiments are provided. If there are any unclear aspects, please feel free to ask further questions.

**Motivation of the Augmentation related equation (Eq10 and Eq 13)**

**Response from the authors**: As the comment from Reviewers p6Bz, a large pool of augmentations and averaging them would give more robust evaluation. In Eq. 10, we use the mean entropy derived from some data augmentation to represent the confidence of the two prior models, and mix up the two priors with a modulating factor. Each addition item is a simple softmax with one type of augmentation, and $\mathcal{I}$ denotes the augmentation types. The softmax is to given the confidence of the source model and $1-\alpha$ means the confidence of the teacher model. Eq. 13 is an improved version for teacher log-likelihood in Eq. 9, which picks up the augmented logits with a larger confidence than the raw data with $\epsilon$ margin. We have conducted the ablation study on augmentation in Appendix E.2, Table 9. Table 9 shows that increasing the number of augmentations can enhance effectiveness.

**Some Typo mistakes**

**Response from the authors**: Reviewers p6Bz, 9v84, and yBx1 all found the paper to be well-written and enjoyable to read. However, they also noted some typographical and grammatical errors, as well as a few confusing statements. Thank you for your careful reviews. We will thoroughly check the paper for any errors and make revisions to enhance clarity and readability. Additionally, we will include more references to support the accuracy of our claims.

Lastly, the authors would like to thank all the reviewers for their diligent and responsible review, as well as for providing high-quality feedback. We believe that the quality of our paper will be significantly improved thanks to your suggestions! If you have any further questions, we welcome continued discussion!

Best regards,

The authors

---

### Decision · Program_Chairs · 2024-09-25

**Decision:**

Reject

**Comment:**

This paper presents a variational Bayesian approach to the problem of continual test-time adaptation (TTA). The proposed approach is inspired by recent works on Bayesian TTA [60], with additional measures to handle the issue of catastrophic forgetting in continual settings. These additional measures are based on recent works on continual TTA [51].

The reviewers appreciate the work for taking a Bayesian approach for continual TTA. However, these were also concerns regarding experiments, e.g., no experiments on other tasks such as segmentation, lack of standard errors in the reported results, and lack of comparison with recent SOTA methods. For the last point, in the rebuttal, although the authors did provide some preliminary numbers (on one dataset, and only for accuracy metric) comparing with EcoTTA, a more detailed experimental comparison is desirable.

In the end, the paper remained somewhat on the borderline. The authors' responses (to the reviewers as well as private comments to the AC) were considered but concerns still remain. The Bayesian setup by itself is not totally a novel idea for TTA (though not for the continual setting, [60] did consider Bayesian TTA, whose formulation is somewhat similar to the proposed method, and then PETAL [2] also had a probabilistic approach to continual TTA) and, given that there is extensive amount of work on continual TTA, the paper also fell short on reporting more extensive comparisons with SOTA methods. I think with the additional comparions with such methods (on multiple datasets, and for classification as well as segmentation, as is routinely done in most continual TTA papers nowadays), it would be a much stronger paper.

The authors are advised to incorporate these suggestions and resubmit the work to another venue.